# Energy-Efficient and Trust-Based Autonomous Underwater Vehicle Scheme for 6G-Enabled Internet of Underwater Things

**DOI:** 10.3390/s25010286

**Published:** 2025-01-06

**Authors:** Altaf Hussain, Shuaiyong Li, Tariq Hussain, Razaz Waheeb Attar, Ahmed Alhomoud, Reem Alsagri, Khalid Zaman

**Affiliations:** 1School of Computer Science and Technology, Chongqing University of Posts and Telecommunications, Chongqing 400065, China; altafkfm74@gmail.com; 2Key Laboratory of Industrial Internet of Things and Networked Control, Ministry of Education, Chongqing University of Posts and Telecommunications, Chongqing 400065, China; 3School of Computer Science and Technology, Zhejiang Gongshang University, Hangzhou 310018, China; 4Management Department, College of Business Administration, Princess Nourah bint Abdulrahman University, P.O. Box 84428, Riyadh 11671, Saudi Arabia; raattar@pnu.edu.sa; 5Department of Computer Science, College of Science, Northern Border University, Arar 91431, Saudi Arabia; aalhomoud@nbu.edu.sa; 6Department of Software Engineering, College of Computer Science and Engineering, University of Hafr Al Batin, Hafr Al Batin 39524, Saudi Arabia; raalsagri@uhb.edu.sa; 7Institute of Intelligent Manufacturing Technology, Shenzhen Polytechnic University, Shenzhen 518000, China; khalidzaman@szpu.edu.cn

**Keywords:** Internet of Underwater Things, autonomous underwater vehicles, 6G-enabled UASN, localization, void node, trust management, EETAUV

## Abstract

This paper introduces a novel energy-efficient lightweight, void hole avoidance, localization, and trust-based scheme, termed as Energy-Efficient and Trust-based Autonomous Underwater Vehicle (EETAUV) protocol designed for 6G-enabled underwater acoustic sensor networks (UASNs). The proposed scheme addresses key challenges in UASNs, such as energy consumption, network stability, and data security. It integrates a trust management framework that enhances communication security through node identification and verification mechanisms utilizing normal and phantom nodes. Furthermore, a 6G communication module is deployed to reduce network delay and enhance packet delivery, contributing to more efficient data transmission. Leveraging Autonomous Underwater Vehicles (AUVs), the EETAUV protocol offers a lightweight approach for node discovery, identification, and verification while ensuring a high data transmission rate through a risk-aware strategy including at low computational cost. The protocol’s performance is evaluated through extensive simulations and compared against state-of-the-art methods across various metrics, including network lifetime, throughput, residual energy, packet delivery ratio, mean square error, routing overhead, path loss, network delay, trust, distance, velocity, Computational Cost of Routing, and data security. The results demonstrate the superior cumulative performance of the proposed EETAUV scheme, making it a robust solution for secure, efficient, and reliable communication in UASNs.

## 1. Introduction

Underwater acoustic sensor networks (UASNs) are part of Underwater Wireless Sensor Networks (UWSNs) which are collections of multiple sensor nodes that operate in underwater regions using acoustic signals and gather data and other meaningful information [1]. After gathering the data, they forward it to the sink nodes on the surface of the water and then the sink nodes transfer it forward to the mobile stations and offshore stations/Base Station (BS) [2]. The communication scenario of Figure 1 is regarding the IoT- and 6G-enabled sensors communication scenario of AUVs, whereas Figure 2 illustrates the working procedure of UASNs. Along with these sensor nodes, there are other advanced nodes called Autonomous Underwater Vehicles (AUVs) which operate using the concept of Internet of Underwater Vehicles (IoUV) and Internet of Underwater Things (IoUT) [3]. Each node is equipped with the ability to move and operate autonomously without the intervention of the BS. These nodes gather real-time information from different areas and are used depending on the application [4]. These nodes have been deployed for specific operations to sense the events and activity underwater and transmit data to the sink nodes [5]. The offshore and onshore BSs are connected directly with the surface sink nodes to obtain meaningful information from the depth sensor nodes [6]. There are numerous dynamic routing strategies and topologies that use fast depletion of energy and high computation cost which needs to be fairly solved with efficient approaches to balance the computational power and battery depletion [7]. Due to some constraints such as the current of water and persistent resistance in the network, frequent disconnection occurs, and with the re-flooding, there are some major research concerns [8]. The IoUT technology, on the other hand, has revolutionized and also expanded the scalability of the network with the connectivity of a large number of devices such as automobiles [9]. Thus, it makes the communication process easy, simple, smarter, and safer with the utilization of 5G/6G communication technologies [10,11]. The 6G network technology has the ability to provide support with high speed and technical standards to meet the coming issues and challenges to provide transmission of big data along with best resource management and efficient communication standards [12,13]. On ocean beds, the deployment and placement of the IoUT devices have achieved outcomes in the delivery of novel technology named IoUV which operates autonomously [14]. Along with the distribution of the traffic load between routes, these devices also control the routing process and improve underwater implementation [15,16].

Due to significant advancements in computing and communication technologies, there is now the potential to enable the deployment of the IoUV on ocean beds [17]. The term IoUV is a more advanced and sophisticated form of the traditional UASNs, in which the primary goal is to ensure simple, fast, and safe autonomous driving in underwater regions [18]. The inclusion and extension of the IoUT systems, internet-enabled services, and inter connected equipment, are accelerating and their applications to transport the networks have been considered innovative [19]. But, with the ongoing trend and growth in the IoUT, it also has a dark side too due to some restricted constraints, like the depletion of energy and security has much more attention among explorers and researchers which offers solutions [20]. On the other hand, numerous IoUT systems are under deployment which include, depending on data from multiple sources, cooperative autonomous driving applications using sensor technologies, safety-critical, management, optimization, and traffic planning [21]. Due to the quality of the shared data and size, the effectiveness of these services is greatly dependent. But, these data are highly sensitive to authenticity concerns and privacy because of their substantial amount. Thus, securing the IoUT is an additional significant and major factor to avoid data compromising and integrity for real-time data. Developing the autonomous system for IoUT remains a significant and challenging task which includes computational intelligence decision and security [22,23]. Making a routing scheme and utilizing smart sensor technologies for autonomous decision making is the core aim of the proposed article. The primary contributions of the proposed EETAUV scheme are summarized as follows:Novel Energy-Efficient and Secure Communication Framework: We propose a 6G-enabled, energy-efficient, and trust-based Autonomous Underwater Vehicle (EETAUV) scheme for UASNs that integrates void node avoidance, localization techniques, and secure communication using normal and phantom nodes for node identification and verification.Enhanced Network Performance and Security: The scheme improves network stability, minimizes delay, increases packet delivery, and ensures secure data transmission with a lightweight, risk-aware strategy supported by AUVs for node discovery and verification.Comprehensive Evaluation: We extensively test the proposed EETAUV scheme against state-of-the-art methods using simulation metrics such as network lifetime, throughput, residual energy, packet delivery ratio, mean square error, routing overhead, path loss, network delay, trust, distance, velocity, computational cost, and data security, demonstrating superior cumulative performance.

This article is organized into the following subsections: The related work is presented in Section 2, and the proposed protocol is described. In Section 3, the simulation environments are covered. In Section 4, the explanations of the experimental results are presented. At last, Section 5 concludes the proposed work.

## 2. Literature Review

The evolution of UASNs and the IoUT has witnessed substantial progress, driven by advancements in communication technologies like 5G and the integration of trust management models. Nkenyereye et al. [1] provided a comprehensive survey on simulation tools for 5G-based underwater networks, laying the foundation for IoUT research. Albekairi [2] introduced a mutable analytics approach to sensor data classification, emphasizing the security and adaptability required for IoUT systems. The energy consumption challenge, critical in UASNs, has been well explored by Mohamed et al. [3], who reviewed energy-efficient routing protocols in wireless sensor networks. The work by Consul et al. [4] built on this by employing deep reinforcement learning to enhance reliable data transmission in 5G-based IoUT networks. Meanwhile, Panicker and Ravi [5] explored high-speed underwater laser communication, pushing the limits of data transmission rates in IoUT, and Massari et al. [6] emphasized the role of AUVs in enabling underwater IoT. These studies collectively reflect the ongoing efforts to optimize energy efficiency and reliability in UASNs. Zhang et al. [7] proposed an energy-aware AUV-assisted data collection strategy, a critical component of sustainable underwater networking. Other notable works, such as Chen et al. [8] and Pirbhulal et al. [9], highlighted the importance of routing and mobility-enabled security for optimizing IoT-based applications. Xu et al. [10] further expanded on multi-AUV pursuit-evasion games in IoUT, demonstrating innovative ways to optimize AUV behavior using reinforcement learning. Trust management, a crucial element for securing UASNs, has been thoroughly investigated by Zhu et al. [11], Jiang et al. [12], and Du et al. [14], each proposing models for identifying malicious attacks, improving trust prediction, and ensuring secure communication. The hybrid trust models and multi-layered security architectures emerging from these studies showcase the need for robust, adaptive frameworks in the dynamic underwater environment. In parallel, the work by Wang et al. [16] and Liu et al. [17] leveraged machine learning algorithms like deep reinforcement learning and random forests to further strengthen trust management and anomaly detection. Zhang et al. [18] introduced a recommendation-based defense mechanism, pushing the boundaries of security in UASNs. Additionally, Zhang et al. [19] and Saeed et al. [20] contributed secure routing protocols like MO-CBACORP and SEECR, prioritizing energy efficiency and cooperation in underwater sensor networks. Muthukkumar and Manimegalai [21] explored dynamic Bayesian games for secured transmission, while Arifeen et al. [22] utilized Hidden Markov Models to manage trust. These trust models, combined with the efforts of Jiang et al. [23] laid the groundwork for future developments in UASNs. Hussain et al. and Shah et al. [24,25] proposed routing schemes CR-NBEER and NBEER for UWSN with a focus on energy efficiency, routing, link stability, path loss, void hole, relay approach, and nearest node identification and verification. Zhu et al. [26] The use of traffic-aware and adaptive routing, as demonstrated by Zhu et al. [27], highlights the importance of real-time decision making in trust management and path reliability. Tomović et al. [28] introduced advanced routing protocols such as DESLR and BEKMP, which incorporate layered routing and blockchain technology to secure communication in UASNs [29]. Similarly, Han et al. [30] applied fuzzy logic to further enhance trust-aware routing. More recent developments in clustering and lifetime maximization strategies have been proposed by [31,32] and Khan et al. [33], addressing network longevity concerns in harsh underwater environments. Ali et al. [34] built on these efforts by proposing the Depth-based Stable Election Routing Protocol, focusing on energy efficiency and heterogeneous IoUT systems.

The evolution of UASNs and IoUT has seen substantial progress, but several challenges persist in the existing works. Energy consumption remains a critical bottleneck for prolonged network performance and operational stability, as highlighted by Mohamed et al. [3], who reviewed energy-efficient routing protocols in wireless sensor networks. While advancements in 5G-based technologies have introduced improvements, issues like high latency, limited bandwidth, and suboptimal routing strategies still hinder real-time underwater communication, as noted by Nkenyereye et al. [1]. Trust management models, extensively studied by Zhu et al. [11], Jiang et al. [12], and Du et al. [14], expose vulnerabilities in trust prediction accuracy and malicious attack identification, which compromise network security. Despite efforts to improve routing, protocols such as MO-CBACORP and SEECR introduced by Zhang et al. [19] and Saeed et al. [20] often fail to balance energy efficiency with reliable communication, particularly in highly dynamic underwater environments.

The integration of AUVs for data collection and mobility-enabled security has demonstrated potential; however, these systems struggle with void node issues, path reliability, and data transmission delays, as highlighted by Zhang et al. [7]. Emerging approaches leveraging machine learning and reinforcement learning, like those by Wang et al. [16] and Liu et al. [17], are computationally intensive, making them unsuitable for lightweight underwater networks. Additionally, advanced routing protocols such as DESLR and BEKMP, proposed by Zhu et al. [26] and Tomović et al. [28], incorporate layered routing and blockchain technologies but face challenges in scalability, adaptability, and sustained network longevity. Lastly, while approaches like the Depth-based Stable Election Routing Protocol by Ali et al. [34] focus on energy efficiency, they lack robust mechanisms for addressing heterogeneous IoUT systems and long-term stability in harsh underwater environments.

The proposed EETAUV scheme addresses the limitations of the existing works by providing a comprehensive solution for UASNs. First, the EETAUV integrates a novel void node avoidance and localization mechanism enabled by 6G communication, ensuring real-time data transmission with minimal delays and improved energy efficiency. It effectively utilizes 6G’s capabilities, such as ultra-low latency, high bandwidth, and enhanced connectivity, to optimize packet delivery while mitigating void node issues. Second, the scheme incorporates a trust-based communication model that secures data transmission through node identification and verification using phantom and normal nodes, thus enhancing network reliability and safeguarding against malicious attacks. This lightweight trust management strategy ensures secure communication while maintaining low computational overhead. Finally, the EETAUV scheme achieves superior network performance and stability by addressing critical metrics like energy consumption, network lifetime, packet delivery ratio, and routing overhead. Through extensive simulations, it demonstrates robust performance compared to state-of-the-art protocols, making it a scalable and adaptive solution for UASNs in dynamic underwater environments.

## 3. Methodology

In this section, we present a comprehensive explanation of the EETUAV scheme, designed to enhance the efficiency and security of routing protocols for autonomous underwater systems. We discuss its developed components in detail. One of the primary challenges faced by autonomous mobile vehicles operating in UASNs is the dynamic flux of moving nodes that interact in real-time. The performance of these autonomous systems heavily relies on efficient data delivery among intercommunicating vehicles while ensuring robust security concerning authentication, integrity, and confidentiality. The proposed EETUAV protocol comprises two main components. The first component addresses the routing of messages from the source to the destination by employing an artificial intelligence-based metaheuristic technique, specifically utilizing a simulated annealing optimization approach. This technique provides flexibility in navigating complex underwater environments and facilitates the efficient forwarding of crucial routing messages to connected autonomous devices. The second component focuses on ensuring the protection and security of communicating devices within the network through session-based authentication procedures. By integrating 6G technology at the upper tier between network edges and sink nodes, the EETUAV scheme effectively explores and overcomes the communication limitations faced by distributed applications in UASNs. This dual approach not only enhances the overall performance of the autonomous system but also instills a high level of trust and reliability in the routing process, critical for the demanding conditions of underwater environments.

### 3.1. System Outlines

This model includes contributions from EETAUV in terms of energy efficiency, void node avoidance, trust, and localization. The simulation requirements and parameters specified are also integrated. The EETAUV scheme consists of the following core parameters.
Sensor Nodes SNs: A total of 500 UASN sensor nodes (SNs).Phantom Nodes: A total of 20 to aid in localization and routing for security and trust management.Key Parameters: Network lifetime (NetLT, dead nodes), throughput (TpT kbps), residual energy (RE, joules), packet delivery ratio (PDR, %), mean square error (MSE), routing overhead (RO, %), path loss (PL dB), network delay (ND ms), trust (P 0&1), distance (m), velocity (m/s), Computational Cost of Routing (CCR %), and data security (%).

### 3.2. Mathematical Model

#### 3.2.1. Energy Consumption Model

The total energy consumed by the EETAUV in a single communication can be expressed as follows:(1)$$E_{EETAUV}=E_{tx}+E_{rx}+E_{proc}+E_{localization}+E_{trust}$$
where $E_{tx}$ = energy consumed in transmission, $E_{rx}$ = energy consumed in reception, $E_{proc}$ = energy consumed in processing, $E_{localization}$ = energy consumed in localization, and $E_{trust}$ = energy consumed in trust assessment. The total energy can also be limited by the total available energy:(2)$$E_{TOTAL}=10_{joules}$$

#### 3.2.2. Energy Components

Transmission Energy
(3)$$E_{tx}=P_{tx}\cdot T_{tx}$$
where $P_{tx}$ = transmission power (*W*), and $T_{tx}$ = transmission time (*s*).Reception Energy
(4)$$E_{rx}=P_{rx}\cdot T_{rx}$$
where $P_{rx}$ = reception power (*W*), and $T_{rx}$ = reception time (*s*).Processing Energy
(5)$$E_{proc}=P_{proc}\cdot T_{proc}$$
where $P_{proc}$ = processing power (*W*), and $T_{proc}$ = processing time (*s*).Localization Energy
(6)$$E_{localization}=P_{localization}\cdot T_{localization}$$
where $P_{localization}$ = power consumed for localization, and $T_{localization}$ = time for localization processes.Trust Assessment Energy
(7)$$E_{trust}=P_{trust}\cdot T_{trust}$$
where $P_{trust}$ = power for trust calculation, and $T_{trust}$ = time for trust assessment.

#### 3.2.3. Distance and Velocity

Assume the number of *AUVs* in a communication region is given by ${AUV}_{i}=\left\{{AUV}_{1},\;{AUV}_{2},\ldots ,\;{AUV}_{3}\right\},$ the distance of the vehicle ${AUV}_{i}$ from the sink node is updated as follows:(8)$$\Delta D_{i}=D_{i}+{AUV}_{v_{i}}$$
where $D_{i}$ is the distance of the moving vehicle ${AUV}_{i}$ from the sink node, and $v_{i}$ is the initial velocity of the vehicle ${AUV}_{i}$.

The change in velocity of the vehicle ${AUV}_{i}$ is given by the following:(9)$$A_{i}={AUV}_{vf}+A{UV}_{vi}\cdot T$$
where ${AUV}_{vf}$ refers to the final velocity of the *AUV*, ${AUV}_{vi}$ refers to the initial velocity of the *AUV*, and T is the time of change in the velocity.

#### 3.2.4. Cost of Routing

The total cost of the route ${CAUVR}_{i}$ from vehicle ${AUV}_{i}$ to ${AUV}_{j}$ can be formulated as:(10)$$C_{{AUVR}_{i}}=\sum_{n=1}^{N}\left(D_{i}+A_{i}\right)$$

To assess the change in the cost of each route, the following equation is applied:(11)$$\Delta AUVC={AUVCR}_{j}-{AUVCR}_{i}$$

If the change in the route cost *ΔAUVC* is less than a minimum threshold, then the new route ${AUVCR}_{j}$ is selected for message routing. The routing cost criteria can be expressed as follows:(12)$$\Delta AUVCeT<rand\left(0,1\right)$$
where *rand*() is a random function ranging from 0 to 1.

### 3.3. Void Node Avoidance Model

To account for void node avoidance, the EETAUV can apply a void detection algorithm based on the residual energy of neighboring nodes. The avoidance can be modeled as follows:(13)$$E_{void}=\sum_{j=1}^{n}E_{residual}\left(j\right)$$
where $E_{residual}\left(j\right)$ = residual energy of neighboring node *j*. If $E_{void}$ < *θ* (threshold energy), switch to an alternate path.

### 3.4. Localization Model

For localization, consider using the Time of Arrival (*ToA*) or Time Difference of Arrival (*TDoA*) methods. The localization energy consumption can be given as follows:(14)$$E_{localization}=\frac{d_{max}}{v_{sound}}\cdot P_{localization}$$
where $d_{max}$ = maximum distance for localization, and $v_{sound}$ = speed of sound in water.

### 3.5. Performance Metrics

Network Lifetime (NetLT*_EETAUV_*)

NetLT*_EETAUV_* depends on the residual energy of sensor nodes and the energy consumption during communication. Along with consumed energy, it can also be measured in the total number of nodes and the number of dead nodes in the network. Mathematically, it can be expressed as follows:(15)$${NetLT}_{EEUAUV}=\frac{E_{total\;energy+nodes}}{\sum_{i=1}^{N}P_{i}(t)}$$
where *E_total_* is the total initial energy consumed and nodes in the network, *Pi*(*t*) is the power consumption of the *i*^th^ node at time *t*, and *N* is the total number of nodes.

EETAUV Contribution: It optimizes energy consumption *Pi*(*t*) via trust-based routing and adaptive communication strategies, improving the network lifetime.

2.Throughput (${TpT}_{EETAUV}$)

${TpT}_{EETAUV}$ in EETAUV is the total number of successfully delivered packets per unit time. It is given as follows:(16)$${TpT}_{EETAUV}=\frac{\sum_{i=1}^{N}D_{i}}{Total_{\_time}}$$
where *Di* is the total data transmitted by node *i* and *total* is the total time for data transmission.

EETAUV Contribution: The scheme enhances throughput by minimizing routing overhead and energy consumption, thus allowing more efficient data transmission.

3.Residual Energy (*RE_EETAUV_*)

RE*_EETAUV_* at any time *t* is the initial energy minus the consumed energy during transmission and reception:(17)$${RE}_{EETAUV}=E_{initial}\left(i\right)-\sum_{i=1}^{t_{max}}P(t)$$
where *E_initial_* is the initial energy of the node, and *P(t)* is the power consumed at time *t*.

EETAUV Contribution: By integrating energy-efficient trust mechanisms, EETAUV reduces power consumption and preserves residual energy.

4.Packet Delivery Ratio (*PDR_EETAUV_*)

*PDR* is the ratio of successfully delivered packets to the total packets sent:(18)$${PDR}_{EETAUV}=\left(\frac{\sum_{i=1}^{N}P_{delivered(i)}}{\sum_{i=1}^{N}P_{sent(i)}}\right)$$
where *P_delivered_*_(*i*)_ is the number of packets successfully delivered by node *i* and *P_sent_*_(*i*)_ is the number of packets sent by node i.

EETAUV Contribution: Trust-based routing increases packet delivery by choosing reliable paths and reducing packet loss due to malicious nodes.

5.Mean Square Error (*MSE_EETAUV_*)

In localization and routing, *MSE* represents the deviation of estimated positions or decisions from the true values. It can be modeled as follows:(19)$${MSE}_{EETAUV}=\frac{1}{N}\sum_{i=1}^{N}(d_{expected}(i)-d_{received}{\left(i)\right)}^{2}$$
where *d_expected_*(*i*) is the expected data for node *i* and *d_received_*(*i*) is the actual data received.

EETAUV Contribution: EETAUV reduces MSE by leveraging trust management to ensure data integrity and mitigate transmission errors.

6.Routing Overhead (*RO_EETAUV_*)

Routing overhead is the percentage of extra packets (control and signaling) transmitted for routing compared to data packets. It is given by the following:(20)$${RO}_{EETAUV}=\left(\frac{\sum_{i=1}^{N}C_{control}(i)}{\sum_{i=1}^{N}D_{data}(i)}\right)$$
where *C_control_*(*i*) represents the control messages generated by node *i*, and *D_data_*(*i*) is the data transmitted by node i.

EETAUV Contribution: EETAUV minimizes routing overhead by optimizing route discovery and maintenance, utilizing trust metrics for efficient communication.

7.Path Loss (*PL****_EETAUV_***)

PL*_EETAUV_* describes the reduction in signal strength as it propagates through the underwater environment.
(21)$${PL}_{EETAUV}\left(dB\right)=log\;10\;\left(\frac{P_{transmitted}}{P_{received}}\right)$$
where *P_transmitted_* is the transmitted signal power, and *P_received_* is the received signal power.

EETAUV Contribution: EETAUV accounts for environmental factors like absorption and scattering to adjust transmission power and mitigate path loss.

8.Network Delay (*ND_EETAUV_*)

ND*_EETAUV_* is the sum of all the delays in packet transmission, queuing, processing, and propagation and can be expressed as follows:(22)$${ND}_{EETAUV}=\frac{{{\sum_{i=1}^{N}\;T}_{transmit}\left(i\right)+T_{process}\left(i\right)+T}_{queue}\left(i\right)}{N}$$
where *t_transmit_*(*i*) is the transmission time for node *i*, *t_process_*(*i*) is the processing time, and *tqueue*(*i*) is the queuing delay.

EETAUV Contribution: EETAUV reduces network delay by optimizing route selection and minimizing communication overhead.

9.Trust (*Tr_EETAUV_*)

Tr*_EETAUV_* measures the trust, security, verification, and reliability of each node in participating in network communication
(23)$${Tr}_{EETAUV}\left(i\right)=\frac{\alpha T_{direct}\left(i\right)+\beta T_{indirect}\left(i\right)}{\alpha +\beta }$$
where *T_direct_*(*i*) is the direct trust of node *i*, *T_indirect_*(*i*) is the indirect trust obtained from neighboring nodes, and α and β are the weight factors.

EETAUV Contribution: EETAUV employs trust metrics to improve routing security and reliability using both direct and indirect trust evaluations.

10.Distance (*d_EETAUV_*)

The distance between communicating nodes i and j influences energy consumption and communication quality is given as follows:(24)$$d_{EETAUV}(i,j)=\sqrt{(x_{i}-x_{j}{)}^{2}+(y_{i}-y_{j}{)}^{2}+(z_{i}-z_{j}{)}^{2}}$$
where *x*_*i*,*j*_, *y*_*i*,*j*_, and *z*_*i*,*j*_ are the coordinates of node *i* and *j*.

EETAUV Contribution: EETAUV optimizes routing decisions based on the physical distance between nodes to reduce energy usage.

11.Velocity (*V_EETAUV_*)

The velocity of the underwater autonomous vehicle affects data transmission timing and path changes.
(25)$$V_{EETAUV}=\frac{D_{start-end}}{t_{start-end}}$$

EETAUV Contribution: EETAUV adapts to dynamic network conditions caused by node mobility to maintain optimal routes.

12.Computational Cost of Routing (*CCR_EETAUV_*)

This metric measures the computational complexity required for routing decisions.
(26)$${CCR}_{EETAUV}=O\left(f\left(N,E,Tr\right)\right)$$
where *f(N,E,Tr)* represents the function depending on the number of nodes *N*, energy *E*, and trust *Tr*.

EETAUV Contribution: EETAUV reduces computational costs by simplifying trust-based decision making and leveraging efficient algorithms for route discovery. The EETAUV scheme optimizes various network metrics such as energy efficiency, trust, and routing overhead, improving the overall network performance in 6G-enabled UASNs.

Figure 3 and Algorithm 1 illustrate the data forwarding scheme of the proposed EETAUV routing process.
**Algorithm 1:** Data forwarding for EETAUV schemeSTART EETAUV_Scheme % Initialize system parametersSET num_nodes = Random(100, 500)SET phantom_nodes = 20SET total_energy = 10 J % Initialize node attributesFOR each node i in num_nodes DOSET node[i].energy = total_energySET node[i].position = Initialize_Position()SET node[i].trust = Initialize_Trust()END FOR% Main LoopWHILE simulation_running DO % Update distances and velocitiesFOR each node i in num_nodes DOnode[i].distance = Update_Distance(node[i])node[i].velocity = Update_Velocity(node[i])END FOR% Check for void nodesFOR each node i in num_nodes DOIF Check_Void_Node(node[i]) THENnode[i].path = Switch_To_Alternate_Path(node[i])END IFEND FOR% Energy Consumption Calculationtotal_energy_consumed = 0FOR each node i in num_nodes DOtotal_energy_consumed += Calculate_Energy_Consumption(node[i])END FOR% Update Trust ValuesFOR each node i in num_nodes DOnode[i].trust = Update_Trust(node[i])END FOR% LocalizationFOR each node i in num_nodes DOlocalization_energy = Calculate_Localization_Energy(node[i])total_energy_consumed += localization_energyEND FOR% Update Total EnergyFOR each node i in num_nodes DOnode[i].energy -= total_energy_consumedEND FOR% Check for network metricsmetrics.network_delay = Calculate_Network_Delay()metrics.pdr = Calculate_PDR()metrics.path_loss = Calculate_Path_Loss()metrics.routing_overhead = Calculate_Routing_Overhead()metrics.mse = Calculate_MSE()metrics.throughput = Calculate_Throughput()% Check energy thresholdsIF total_energy_consumed > total_energy THENsimulation_running = FALSEEND IFEND WHILE% Output Performance MetricsOutput(metrics)END EETAUV_Scheme

The flowchart illustrates the step-by-step mechanism of data forwarding and routing in the EETAUV protocol to ensure energy-efficient and secure communication in 6G-enabled UASNs. The process begins with the initialization phase, where the route to the *AUVR* is established. This initialization leverages 6G communication to enable high-speed and low-latency data transfer, crucial for dynamic underwater environments. At this stage, the route cost (*AUVR*) is calculated to determine the most optimal communication path.

Next, the system integrates energy-efficient metrics to enhance the overall network lifetime by minimizing energy consumption during communication. This is critical for underwater networks, where energy resources are limited. The protocol then checks for void nodes—nodes that are either unreachable or have insufficient energy. If a void node is detected, the protocol switches to an alternate path and selects a neighboring route (*AUVRj*). The newly selected route undergoes a validity check to ensure it is suitable for forwarding data. If no valid node is found, the system updates the current route to maintain network stability.

When no void node is detected, the system transitions to the localization and 6G parameter phase, where the route cost is recalculated using advanced localization techniques and updated 6G communication parameters. These updates ensure that the routing path remains efficient and reliable. The protocol then evaluates the change in route cost (Δ*AUVRC*). If the change surpasses a predefined threshold, the system applies the Memphis criterion to verify the route based on 6G-specific thresholds. This step ensures that the routing path adheres to quality standards and performance expectations.

Finally, once all the conditions are satisfied, the current route is updated to reflect the optimized routing path. This continuous process guarantees optimal network performance, energy efficiency, and reliable communication. The flowchart concludes when a stable route is achieved, or if further updates are unnecessary. This structured approach highlights the robustness and adaptability of the EETAUV protocol in addressing energy constraints and maintaining trust in underwater sensor networks.

Algorithm 1 details the inner workings of the EETAUV protocol. It starts by initializing system parameters such as the number of nodes, phantom nodes (for added security), and total energy available for each node. Each node is assigned initial attributes, including position and trust values, which facilitate secure and efficient routing. During the simulation, nodes dynamically update their distances and velocities to adapt to changing underwater environments. Void node detection plays a critical role in maintaining network robustness. If a node is identified as void, it switches to an alternate path to prevent communication breakdowns.

Energy consumption is carefully monitored throughout the process. The algorithm calculates the total energy consumed by all the nodes, updating their trust values periodically to maintain secure communication. Localization energy is also calculated, accounting for the energy spent in determining node positions. Nodes’ energy levels are subsequently updated to reflect their consumption. Network performance is evaluated using several key metrics, including network delay, PDR, path loss, routing overhead, MSE, and throughput. These metrics ensure the protocol meets performance standards while conserving energy.

The protocol stops if the total energy consumed exceeds the available energy, signaling the end of the simulation. Otherwise, it continues optimizing routing decisions and network operations. The final output includes performance metrics, providing a comprehensive evaluation of the protocol’s effectiveness. Overall, the EETAUV protocol is robust and efficient, leveraging 6G-enabled features, trust-based mechanisms, and adaptive energy management to support secure and reliable underwater communication. It is particularly suited for dynamic underwater environments where energy efficiency, adaptability, and secure communication are paramount.

Trust-enabled algorithm with data security in 6G-enabled UASNs, focusing on trust management, send and receive acknowledgement, mutual authentication, *RREQ-RREP*, unique *ID*, neighbor node identification, and verification for the proposed EETAUV scheme.

Figure 4 and Algorithm 2 illustrate the trust-based and security procedure of the proposed EETAUV scheme.
**Algorithm 2:** Trust management EETAUV scheme for 6G-enabled UASNsInitialize:Set unique ID for each AUVs and sensor nodesInitialize trust values for every neighbor nodeSet threshold trust value T_thDefine time intervals for sending/receiving acknowledgment (ACK)Define RREQ and RREP for routing requests and repliesStart AUV/Node Communicationfor each node in the network:Broadcast “HELLO” packet with unique ID to neighbor nodesReceive response from neighbor nodesNeighbor Identification and Verification:for each received “HELLO” response:Extract sender’s unique IDVerify unique ID using secure hashing functionif (ID is valid):Add to verified neighbor listelse:Discard node as untrustworthyMutual Authentication:for each verified neighbor:Send Mutual Authentication Request (MAR) with encrypted keysReceive Mutual Authentication Response (MARP)if (MARP is valid):Update neighbor node’s trust valueelse:Mark as untrustworthyTrust Management:for each verified node:Calculate trust value based on:Successful packet transmissions (T_s)Acknowledgments received (T_ack)Response to RREQ (T_rreq)Data integrity (T_data)Trust_value = w1 * T_s + w2 * T_ack + w3 * T_rreq + w4 * T_dataif (Trust_value < T_th):Mark node as untrustworthyelse:Keep node in trusted neighbor listRoute Discovery (RREQ-RREP):if (destination is unknown):Broadcast RREQ with unique ID and trust levelfor each received RREP:Verify the trust level of the responding nodeif (trust level is higher than T_th):Select route through that nodeelse:Continue route discoveryData Transmission:Select the trusted route based on trust value and energy efficiencySend data packet via the selected trusted routefor each hop:Receive ACK from next hopUpdate trust value based on ACK receptionif (ACK not received within timeout period):Re-initiate route discoveryAcknowledgment and Feedback:if (data packet successfully delivered):Receive final ACK from destinationIncrease trust level of participating nodeselse:Decrease trust level of nodes in the routePeriodic Trust Update:for each time interval:Update trust values for all neighborsRecalculate based on recent communication historyEnd

### 3.6. Trust-Based Model

Trust for EETAUV can be calculated and updated as follows:(27)$$T_{EETAUV}\left(i\right)=\frac{\sum_{j=1}^{n}R_{ij}}{n}$$
where $R_{ij}$ = reputation score from interactions with node *j*.

To update the trust value based on new interactions
(28)$$T_{EETAUV}{\left(i\right)}_{new}=\alpha \cdot T_{EETAUV}{\left(i\right)}_{old}+\left(1-\alpha \right)\cdot R_{new}$$
where *α* denotes the decay factor.

The transition from 3G, 4G, and 5G to 6G in implementing the EETAUV scheme for UASNs reflects a significant evolution in communication technology. Each previous generation faced specific limitations that constrained their effectiveness in underwater and high-demand network environments, particularly in terms of bandwidth, latency, energy efficiency, and adaptability. Third-generation networks, while revolutionary at their time, primarily focused on enabling basic mobile internet services and had limited bandwidth and high latency, making them unsuitable for handling the vast data exchange requirements and real-time communication demands of UASNs. Fourth-generation networks introduced higher speeds and better capacity; however, their coverage and energy efficiency still posed challenges for underwater networks, where signal attenuation and energy constraints are significant hurdles. Furthermore, 4G lacked the ultra-low latency and robust data-handling capabilities required for dynamic underwater scenarios. The fifth generation, with its higher bandwidth, enhanced connectivity, and URLLC, made notable progress, particularly for terrestrial IoT and smart systems. However, even 5G struggled in underwater environments due to its limited ability to adapt to high levels of signal attenuation, dynamic mobility, and energy inefficiencies. Additionally, 5G networks still lacked the computational intelligence and optimization capabilities necessary for addressing the complex routing, localization, and trust management challenges posed by UASNs.

In contrast, 6G-enabled nodes address these issues and are uniquely suited for the EETAUV scheme due to their advanced technological features. Sixth-generation networks are designed to deliver terahertz (THz) frequencies, offering unprecedented bandwidth and ultra-low latency, which are crucial for supporting the high data rates and real-time communication required in underwater environments. This is particularly important for the EETAUV scheme, which relies on dynamic updates of node positions, trust values, and route costs. Moreover, 6G introduces energy-aware communication techniques, such as energy harvesting and adaptive power control, significantly extending the lifetime of underwater sensor nodes and addressing one of the critical limitations of earlier generations. Additionally, 6G incorporates artificial intelligence (AI) and machine learning (ML) frameworks, which enhance the routing and decision-making processes in EETAUV by providing predictive analytics, trust management, and adaptive routing strategies. This makes 6G more reliable in managing void node detection, route optimization, and energy consumption. Furthermore, 6G networks provide ubiquitous coverage, enabling seamless connectivity even in challenging underwater environments where traditional signals experience high path loss and interference. The integration of quantum communication and blockchain technologies in 6G enhances security and trust, addressing vulnerabilities in earlier networks, especially critical for ensuring secure data transmission in UASNs. Lastly, 6G’s ability to handle extreme-scale IoT connectivity supports the high node density in UASNs, ensuring efficient communication and data forwarding even with hundreds of nodes. In short, while 3G, 4G, and 5G made incremental advancements, their limitations in bandwidth, latency, energy efficiency, and adaptability rendered them less suitable for the complex requirements of UASNs. Sixth generation’s superior capabilities, such as ultra-wide bandwidth, AI-driven optimization, energy-efficient mechanisms, and enhanced security, make it the ideal choice for implementing the EETAUV scheme, ensuring optimal performance and reliability in underwater communication networks.

## 4. Experiments

In this section, we outline the simulation environment used to conduct various experiments and compare their results with other solutions. The proposed protocol was implemented in MATLAB, and simulations were executed accordingly. The network dimensions were set to 800 × 500 × 50 m^3^, while for phantom nodes, dimensions were set to 800 × 500 × 200 m^3^ with sensor nodes (SNs) randomly deployed in quantities of 50, 100, 150, and up to 500. The experiments were run for 1000 rounds, measuring network performance across two scenarios: the total number of rounds and the number of SNs. Additionally, the number of edge devices and phantom nodes were configured to 15 and 20, respectively. Table 1 presents the simulation parameters and settings adjusted for the EETUAV scheme, while Figure 5 and Figure 6 illustrate the deployment scenarios of the SNs, phantom nodes, and edge devices.

### 4.1. Results

Here, a detailed discussion of the key evaluation metrics and justification is given as to why EETAUV consistently outperformed well against all the other schemes in UUASNs.

#### 4.1.1. Network Lifetime (NetLT Number of Dead Nodes)

Network lifetime is measured by the number of rounds or nodes until the first node dies. As UASNs operate in harsh underwater environments, maximizing network lifetime is critical to prolong the operational effectiveness of the entire network. In both simulation round- and node-based scenarios, EETAUV outperformed the other protocols by sustaining a higher network lifetime, particularly as simulations progressed. For instance, in the last round of simulations, EETAUV maintained 410 rounds compared to lower values such as 300 and 320 for EECP and GHL-SAR, respectively. This indicates that EETAUV is highly energy-efficient, primarily due to its optimized energy consumption and trust-based communication mechanisms. By reducing unnecessary transmissions and optimizing relay node selection based on trustworthiness, EETAUV prevents premature energy depletion in individual nodes, which is a common cause of network failure in UASNs. The reason behind this improved network lifetime can be attributed to the trust management system and the energy-aware routing embedded in EETAUV. The trust mechanism ensures that only reliable nodes participate in communication, reducing the need for retransmissions, which conserves energy. Additionally, EETAUV’s efficient relay mobility and adaptive power control further contribute to energy savings, leading to enhanced network longevity. Figure 7a,b are regarding the evaluation against simulation rounds and varying numbers of nodes.

#### 4.1.2. Throughput (TpT, kbps)

Throughput measures the rate of successful message delivery over the network, typically represented in kilobits per second (kbps). For UASNs, maintaining high throughput is essential for applications like underwater exploration, surveillance, and monitoring, where large volumes of data must be transmitted in real-time. EETAUV excelled in throughput in both round and node-based evaluations, maintaining values as high as 2000 kbps in the initial rounds and sustaining a strong throughput of 1050 kbps in the final rounds, surpassing competing protocols like T-SAPR and TAFLRLR, which dropped as low as 750 kbps. Similarly, in the node-based evaluations, EETAUV consistently remained superior, ending with a throughput of 1300 kbps for 500 nodes, whereas other protocols like BEKMP and SEP-IoUT struggled to exceed 950 kbps. The reason EETAUV maintains high throughput is largely due to its multi-hop and direct communication mechanism, which minimizes communication delays and maximizes the successful delivery of data packets. The RSSI-based forwarding mechanism optimizes route selection by choosing paths with the strongest signal strength, ensuring that data are transmitted with minimal loss or interference. The adaptive nature of the protocol allows it to dynamically adjust to changes in network topology, ensuring that throughput remains consistently high even as node density or network traffic increases. Figure 8a,b are regarding TpT evaluation against simulation rounds and varying numbers of nodes.

#### 4.1.3. Residual Energy (RE Joules)

Residual energy is a crucial parameter that reflects how well a protocol conserves node energy over time. High residual energy signifies energy-efficient routing, which is particularly important for UASNs, where replacing or recharging node batteries is challenging. EETAUV demonstrated significantly better residual energy retention across the simulation rounds, particularly in the later stages. For example, by round 1000, EETAUV retained an impressive 8.3 joules compared to T-SAPR’s 4.8 joules and EECP’s 3.8 joules. The same trend was observed in the node-based simulations, where EETAUV retained 8.5 joules at the 500-node mark, while other protocols like GHL-SAR and BEKMP exhibited much lower values, barely reaching 6 joules. This high residual energy can be attributed to adaptive power control and optimized communication paths employed by EETAUV. The protocol ensures that only minimal power is used for transmissions by dynamically adjusting transmission power based on node proximity and signal strength. Furthermore, the trust-based algorithm ensures that only the most energy-efficient and trustworthy nodes are involved in the communication process, reducing energy wastage from malicious or unreliable nodes. Figure 9a,b are regarding RE evaluation against simulation rounds and varying numbers of nodes.

#### 4.1.4. Packet Delivery Ratio (PDR %Age)

PDR is the ratio of packets successfully delivered to the destination over the total packets sent. A high PDR is critical in UASNs to ensure data reliability, especially for mission-critical applications such as disaster monitoring and environmental data collection. EETAUV maintained the highest PDR, reaching 100% in the early rounds and gradually reducing to 82% in the final round. In comparison, other protocols like SEP-IoUT and EECP saw steep declines, falling below 60% in the same rounds. The same pattern was observed in the node-based simulations, where EETAUV’s PDR consistently remained around 83% for 500 nodes, while others like BEKMP and GHL-SAR dropped to 45%. The dual-sink anycasting approach utilized by EETAUV is one of the key reasons for this high PDR. This approach allows the protocol to send packets to multiple sinks, increasing the likelihood of successful delivery even in the presence of node or link failures. The use of multi-hop communication also helps ensure that packets are not dropped prematurely, as they are relayed across trusted nodes with optimal energy and signal conditions. Figure 10a,b are regarding PDR evaluation against simulation rounds and varying numbers of nodes.

#### 4.1.5. Mean Square Error (MSE)

MSE measures the accuracy and efficiency of the protocol in predicting or estimating performance metrics like delay, energy consumption, and packet delivery. Lower MSE values are desirable as they reflect better reliability and efficiency. EETAUV consistently exhibited the lowest MSE across both round- and node-based simulations, with values as low as 0.15 in the final rounds compared to 0.42 for SEP-IoUT and 0.33 for BEKMP. In the node-based simulations, EETAUV achieved an MSE of 0.45 with 500 nodes, while others like GHL-SAR struggled to reduce below 0.55. The low MSE in EETAUV is indicative of its superior predictive model and decision-making algorithms. The protocol utilizes a combination of trust management and feedback mechanisms to refine routing decisions and ensure that data are delivered with minimal errors. Additionally, the integration of adaptive path loss mitigation techniques ensures that energy consumption and delay predictions are accurate, further contributing to the reduced MSE values. Figure 11a,b are regarding MSE evaluation against simulation rounds and varying numbers of nodes.

#### 4.1.6. Routing Overhead (RO %Age)

Routing overhead refers to the percentage of additional control packets required for route discovery and maintenance. In UASNs, minimizing routing overhead is crucial for reducing energy consumption and communication delays. When compared with the nine other protocols, EETAUV consistently outperforms by exhibiting the lowest routing overhead. For instance, in the simulation rounds, EETAUV starts with an overhead of 8% and increases gradually to 17%, while other protocols like TECTM and GHL-SAR show significantly higher values, reaching as much as 36% and 33%, respectively, by the end. EETAUV’s better performance can be attributed to its energy-efficient routing mechanism, which intelligently selects routes based on both energy consumption and link quality. By employing relay mobility and adaptive hop-by-hop forwarding, EETAUV minimizes unnecessary control packet transmissions and recalculations of routes, thus keeping overhead low even with an increasing number of nodes and rounds. Additionally, the dual sink design reduces the need for re-routing, as the presence of multiple forwarding paths enhances network reliability and efficiency. This feature sets EETAUV apart from protocols like BEKMP and T-SAPR, which incur higher overhead due to frequent re-routing in the dynamic underwater environment. Figure 12a,b are regarding RO evaluation against simulation rounds and varying numbers of nodes.

#### 4.1.7. Path Loss (PL, dB)

Path loss, measured in decibels (dB), represents the attenuation of signal strength as it propagates through the underwater environment. A high path loss indicates weak signals, which leads to inefficient communication. In both the simulation rounds and node tests, EETAUV consistently maintains lower path loss compared to its counterparts. For example, in the 1000 rounds test, EETAUV’s path loss starts at 75.5 dB and gradually rises to 79.0 dB, while other protocols like EECP and TAFLRLR experience much higher losses, reaching up to 90.1 dB. The superiority of EETAUV in reducing path loss can be explained by its optimized energy consumption model and adaptive power control. EETAUV uses real-time RSSI-based forwarding and considers Line-of-Sight (LoS) and Non-Line-of-Sight (NLoS) adaptation, allowing it to dynamically adjust transmission power to minimize losses during communication. Furthermore, its dual sink anycasting helps improve the reliability of signal transmission by offering multiple paths, thus reducing the likelihood of communication through high-attenuation zones. These techniques ensure that signals are transmitted through the most efficient paths, unlike protocols like SEP-IoUT and EECRAP which fail to adapt effectively to changing underwater conditions. Figure 13a,b are regarding PL evaluation against simulation rounds and varying numbers of nodes.

#### 4.1.8. Network Delay (ND, ms)

Network delay refers to the time it takes for data to traverse from the source to the destination. In UASNs, reducing network delay is critical for time-sensitive applications, such as monitoring marine environments or conducting underwater research. The results indicate that EETAUV performs exceptionally well in minimizing delay, with values ranging from 15.7 ms to 20.8 ms, as seen in the rounds. Comparatively, protocols like BEKMP and TAFLRLR experience delays exceeding 30 ms. EETAUV’s low delay is attributed to its multi-hop communication strategy and the efficient use of motion sink and relay nodes. By using a proactive routing approach that takes node mobility and dynamic topology changes into account, EETAUV is able to quickly adapt and re-route data in case of failures or congestion. Its fast convergence time also reduces the time taken to establish new routes, thereby preventing delays caused by re-transmissions or link failures. The combination of adaptive hop-by-hop forwarding and dual synchronization mechanisms helps in maintaining a smooth and delay-free data flow, making it superior to the other protocols that suffer from higher latencies due to inefficient route maintenance. Figure 14a,b are regarding ND evaluation against simulation rounds and varying numbers of nodes.

#### 4.1.9. Trust (Tr, Probability)

Trust is a measure of the reliability and security of nodes in the network. In UASNs, where malicious nodes or failures can compromise data integrity, maintaining high trust levels is essential. EETAUV scores the highest trust values among all the protocols, starting at 0.81 and reaching 0.95, as shown in the round plot. Other protocols, such as EECRAP and SEP-IoUT, exhibit trust levels that plateau around 0.78–0.84, indicating less reliability. EETAUV’s higher trust values can be attributed to its trust-based algorithm that incorporates mutual authentication and neighbor node verification. By assigning trust scores to nodes based on their behavior (such as packet forwarding accuracy and acknowledgment responses), EETAUV ensures that only trusted nodes participate in data routing. The scheme’s ability to dynamically adjust trust levels based on real-time network behavior prevents malicious nodes from remaining in the network for extended periods. Additionally, the use of secure acknowledgment and unique node ID further strengthens the overall security framework, ensuring that the network remains resilient to attacks or misbehavior, which is not as robust in protocols like DESLR and GHL-SAR. Figure 15a,b are regarding Tr evaluation against simulation rounds and varying numbers of nodes.

#### 4.1.10. Distance (D, m)

Distance between nodes plays a crucial role in UASNs, as it affects the signal propagation time and the overall energy consumption. EETAUV efficiently handles varying distances between nodes, maintaining a balanced communication framework throughout. For instance, the distance values in the simulation rounds show that EETAUV maintains consistent and optimal communication, which prevents excessive delays or path loss that could be caused by long distances. The EETAUV protocol excels in managing distance-related challenges through its mobility-adaptive approach. Its relay nodes are strategically placed to ensure that data packets are forwarded through nodes that are within an optimal range, thus preventing unnecessary energy consumption or packet losses due to long distances. The ability to dynamically adjust the network topology based on node mobility further enhances its performance, as opposed to static routing protocols like T-SAPR and EECP which struggle with maintaining efficient communication over longer distances. Figure 16a,b are regarding Di evaluation against simulation rounds and varying numbers of nodes.

#### 4.1.11. Velocity (V, m/s)

Velocity plays a critical role in underwater autonomous vehicle (UAV) networks due to the fluid and dynamic nature of the underwater environment. Efficient navigation and movement are crucial for reliable communication and network performance. In the tested scenarios, EETAUV consistently demonstrated superior velocity management compared to the other protocols. The values across both rounds and nodes show a steady increase in velocity, peaking at 15.0 m/s and 13.2 m/s, respectively, outperforming the other protocols like BEKMP (max 11.0 m/s in rounds) and T-SAPR (max 10.3 m/s in nodes). This performance can be attributed to the adaptive control mechanisms of EETAUV, which enable optimized speed regulation while maintaining network stability. The use of 6G-enabled communication modules reduces communication delays, allowing autonomous vehicles to move faster without sacrificing data packet integrity. Additionally, the integration of AUVs ensures efficient node discovery, which reduces latency and improves mobility. EETAUV’s trust-based control over velocity, combined with the use of lightweight communication modules and an adaptive movement model, allows for faster yet stable movements in underwater environments. Its ability to maintain higher speeds while balancing energy consumption and data integrity gives it an edge over conventional protocols, which might slow down due to computational overheads and inefficient routing mechanisms. Figure 17a,b are regarding V evaluation against simulation rounds and varying numbers of nodes.

#### 4.1.12. Computational Cost of Routing (CCR %Age)

The CCR is an essential metric that indicates the efficiency of routing algorithms in terms of the processing power and energy required to compute optimal paths. EETAUV exhibits exceptional performance in reducing computational costs, achieving the lowest CCR values compared to all the other protocols. At 1000 simulation rounds, EETAUV maintains a CCR of 43.5%, far below protocols like GHL-SAR and TECTM, which exhibit CCRs of 54.1% and 53.0%, respectively. EETAUV’s low CCR is achieved through its void node avoidance mechanism and trust-based routing decisions, which minimize unnecessary computation by proactively identifying and avoiding areas with void nodes. Additionally, its node identification and verification process, supported by the deployment of phantom and normal nodes, ensures that only secure and reliable routes are computed. This not only reduces the computational burden but also improves the overall network efficiency. The use of trust-aware strategies for routing decisions and the efficient handling of node discovery reduces the need for complex routing recalculations. EETAUV leverages lightweight algorithms for routing computations, making it more energy-efficient and computationally less expensive. By avoiding redundant calculations, it outperforms the other protocols with high computational overheads that struggle with dynamic underwater conditions. Figure 18a,b are regarding CCR evaluation against simulation rounds and varying numbers of nodes.

#### 4.1.13. Data Security (%Age)

Security is a paramount concern in UASNs, where data transmission is prone to eavesdropping, interception, and malicious attacks. EETAUV incorporates robust security mechanisms by using a combination of node identification, verification, and phantom nodes to secure data transmission. The data security percentage for EETAUV remains the highest across all the test rounds and node configurations, maintaining a consistent 100% in the first round and a gradual decrease to only 91% after 1000 rounds. In contrast, protocols like T-SAPR and SEP-IoUT experience a more significant drop in data security over time, reaching as low as 70–75%. EETAUV’s high data security can be attributed to its risk-aware strategy and the implementation of normal and phantom nodes that safeguard communication by preventing unauthorized access. Moreover, the trust-based model ensures that only verified nodes can participate in the communication process, thus enhancing data integrity and confidentiality. EETAUV excels in maintaining high data security due to its dual-layered security framework. The trust-based model, coupled with secure node identification, makes it highly resistant to attacks and data breaches. This advantage is particularly important in UASNs, where environmental challenges make secure communication more difficult. The superior performance of EETAUV compared to the other protocols is a testament to its carefully designed security algorithms, which balance speed, trust, and energy efficiency. Figure 19a,b are regarding data security evaluations against simulation rounds and varying numbers of nodes. Table 2 illustrates the overall performance of all the schemes against each parameter.

### 4.2. Discussion

UASNs present unique challenges due to their harsh and dynamic environments, where factors such as energy efficiency, communication reliability, and data security are critical for the overall performance of the network. The proposed EETAUV scheme offers a robust solution to these challenges by optimizing key network performance parameters across several metrics, including NetLT, TPT, RE, PDR, MSE, RO, PL, ND, trust, velocity, and CCR. Each of these parameters highlights the superiority of EETAUV in comparison to other leading protocols like EECP, GHL-SAR, BEKMP, and SEP-IoUT, particularly in scenarios involving extended simulation rounds and large numbers of sensor nodes. NetLT is one of the most significant metrics for UASNs due to the difficulty of replacing or recharging nodes in underwater environments. EETAUV’s trust-based communication mechanisms, optimized relay node selection, and energy-aware routing have enabled it to outperform other protocols, sustaining network operation for a longer duration. The results show that EETAUV’s trust management system, which ensures that only reliable nodes participate in communication, plays a crucial role in prolonging network lifetime by reducing retransmissions and unnecessary energy consumption. This, combined with efficient relay mobility and adaptive power control, ensures that the energy of individual nodes is conserved, leading to an extended network lifespan. TPT is another vital metric for UASNs, particularly for applications requiring the transmission of large volumes of data in real time. EETAUV excels in this regard by maintaining high throughput levels even as the number of rounds or nodes increases. The protocol’s multi-hop and direct communication mechanism, along with RSSI-based forwarding, ensures that data packets are delivered efficiently with minimal loss. This ability to adapt dynamically to changing network conditions, such as node density or traffic, allows EETAUV to maintain consistently high throughput, making it suitable for real-time underwater applications. RE is a reflection of the energy efficiency of a protocol. In UASNs, where energy conservation is paramount, EETAUV demonstrates superior residual energy retention. This is achieved through the protocol’s adaptive power control and optimized communication paths, ensuring that only the minimal required energy is used for transmissions. The trust-based algorithm employed by EETAUV further enhances energy efficiency by selecting trustworthy and energy-efficient nodes for communication, thereby preventing energy wastage caused by unreliable nodes. PDR is critical in ensuring data reliability in UASNs, especially for mission-critical applications. EETAUV’s dual-sink anycasting approach and multi-hop communication mechanism contribute significantly to its high PDR, ensuring that data packets are successfully delivered even in challenging underwater environments. The ability to send packets to multiple sinks increases the likelihood of successful delivery, making EETAUV a reliable protocol for data-intensive underwater operations. MSE, which measures the accuracy of the protocol’s performance predictions, is another area where EETAUV excels. The low MSE values achieved by EETAUV indicate its superior predictive model and decision-making algorithms. The integration of trust management and feedback mechanisms allows EETAUV to refine its routing decisions continuously, ensuring minimal errors in data transmission. RO% is a key concern in UASNs, as excessive control packet transmissions can lead to increased energy consumption and communication delays. EETAUV significantly reduces routing overhead by employing energy-efficient routing mechanisms that minimize unnecessary control packet transmissions. The dual-sink design and adaptive hop-by-hop forwarding help to maintain low overhead levels even as the network grows in size, setting EETAUV apart from protocols that incur higher overhead due to frequent re-routing. PL, which affects signal strength and communication efficiency, is minimized in EETAUV through its adaptive power control and RSSI-based forwarding mechanisms. By dynamically adjusting transmission power based on the signal strength and the presence of Line-of-Sight (LoS) and Non-Line-of-Sight (NLoS) conditions, EETAUV ensures efficient signal propagation and reduces communication losses. This makes EETAUV particularly effective in managing the challenging propagation conditions found in underwater environments. ND is critical for time-sensitive underwater applications, and EETAUV’s low delay values demonstrate its effectiveness in ensuring fast and reliable data transmission. The protocol’s multi-hop communication strategy, combined with the use of motion sink and relay nodes, ensures that data packets traverse the network quickly, avoiding the delays commonly associated with re-transmissions or link failures in other protocols. Trust is a vital metric in UASNs, where malicious nodes can compromise the integrity of data transmission. EETAUV’s trust-based algorithm ensures high trust levels by incorporating mutual authentication, neighbor node verification, and trust-based decision making in routing. This prevents malicious nodes from disrupting network operations and enhances the overall security and reliability of the network. Velocity management is essential for underwater autonomous vehicle (UAV) networks, where node mobility can affect network stability. EETAUV’s adaptive control mechanisms enable it to maintain high velocity without compromising data integrity or network performance. The use of 6G-enabled communication modules reduces communication delays, allowing for faster movement and efficient node discovery. Finally, CCR is minimized in EETAUV through the use of lightweight algorithms and trust-aware strategies that reduce the need for complex routing recalculations. By avoiding redundant calculations and optimizing routing decisions, EETAUV outperforms other protocols that struggle with high computational overheads in dynamic underwater conditions. In summary, EETAUV offers a comprehensive and efficient solution to the challenges faced by UASNs, excelling across multiple performance metrics. Its trust-based routing, energy-efficient design, and adaptive communication mechanisms make it a highly effective protocol for underwater acoustic sensor networks, particularly in demanding environments where energy conservation, data reliability, and security are critical. Table 3, Table 4 and Table 5 illustrate the overall cumulative and average values of each scheme in terms of scale, rounds, and varying number of sensor nodes.

While the EETAUV scheme offers several advantages in terms of energy efficiency, trust management, and the integration of 6G communications for UASNs, there are some potential drawbacks and areas for improvement. One limitation is the reliance on 6G communication, which, although promising, is still in the early stages of development, and its real-world deployment in underwater environments could face challenges related to infrastructure availability and high implementation costs. Additionally, while the scheme effectively addresses energy consumption and trust management, the complexity of the algorithm may increase computational overhead, particularly in large-scale networks with numerous nodes, potentially leading to delays in real-time decision making. Future work could focus on optimizing the routing algorithm for better scalability and reduced computational cost, as well as exploring hybrid communication models to balance energy efficiency and network reliability. Furthermore, incorporating more adaptive mechanisms for handling environmental changes, such as varying ocean currents and node mobility, could further enhance the robustness of the scheme in dynamic underwater environments.

## 5. Conclusions

In this paper, we proposed the EETAUV protocol for 6G-enabled UASNs, addressing critical issues such as void node avoidance, energy consumption, and security. The proposed scheme effectively enhances network performance and stability by integrating trust management with secure communication mechanisms, including node identification and verification via normal and phantom nodes. By deploying a 6G communication module, EETAUV significantly reduces network delay and improves packet delivery, contributing to optimized data transmission. Additionally, the use of AUVs ensures lightweight node discovery and verification while maintaining high data transmission reliability. Extensive simulations demonstrate that the EETAUV protocol outperforms state-of-the-art solutions across key performance metrics, including network lifetime, throughput, residual energy, packet delivery ratio, and data security. Overall, the proposed protocol proves to be an efficient and secure solution for UASNs, contributing to the advancement of underwater communication technologies. For future research, we plan to explore the integration of machine learning techniques into the EETAUV protocol to enhance the decision-making processes related to node trust management, energy optimization, and communication reliability. Additionally, we aim to investigate adaptive strategies for dynamic environmental changes in UASNs, allowing the protocol to respond effectively to varying underwater conditions. Expanding the scope of the protocol to support multi-AUV collaboration and 3D localization will also be considered to improve scalability and robustness in more complex underwater scenarios. Finally, we will evaluate the real-time implementation of the EETAUV protocol using hardware testbeds to validate its performance in practical underwater environments.

## Figures and Tables

**Figure 1 sensors-25-00286-f001:**
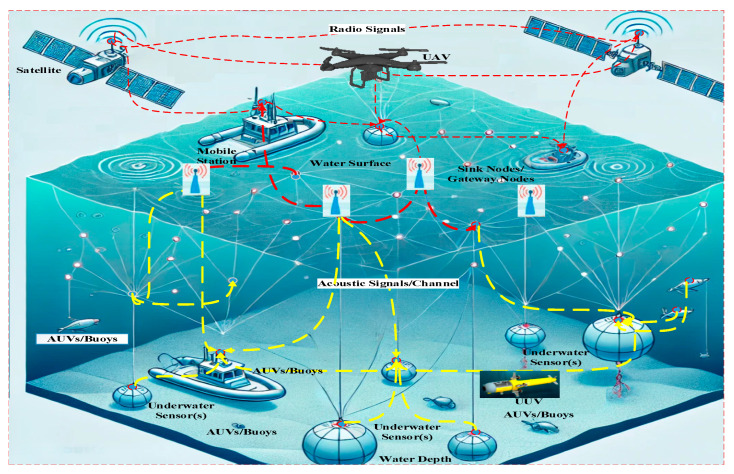
Sixth-generation and IoUT-based AUVs communication scenario.

**Figure 2 sensors-25-00286-f002:**
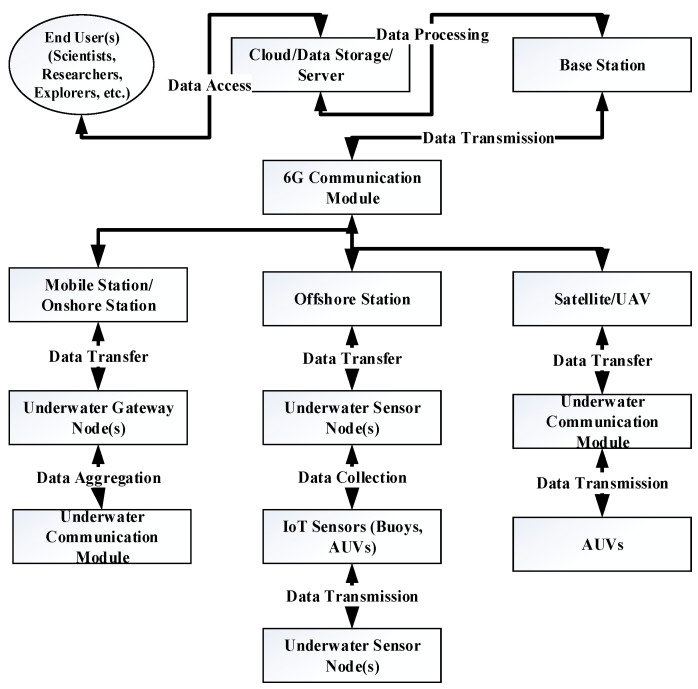
Working procedure of UASN.

**Figure 3 sensors-25-00286-f003:**
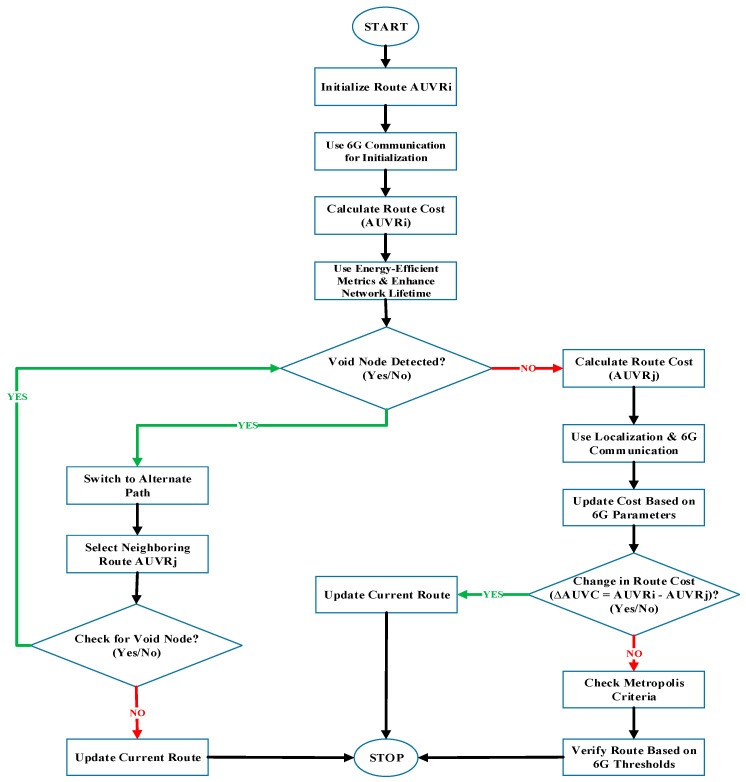
Data forwarding flowchart of the EETAUV scheme.

**Figure 4 sensors-25-00286-f004:**
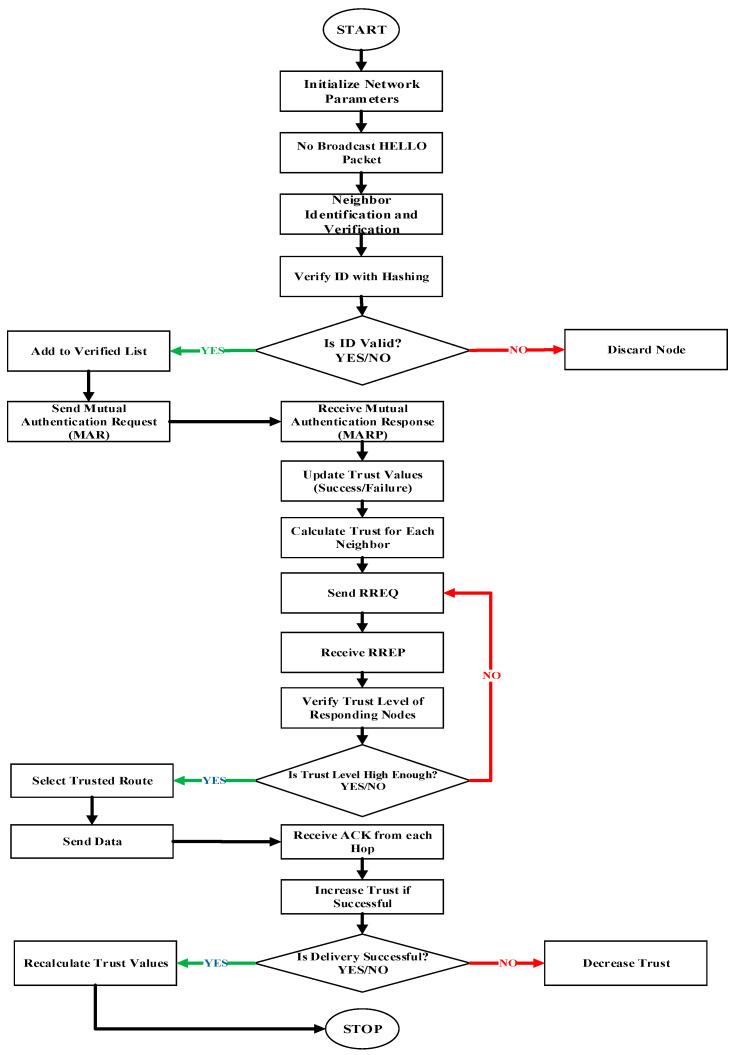
Node identification and verification procedures for trust-based communication.

**Figure 5 sensors-25-00286-f005:**
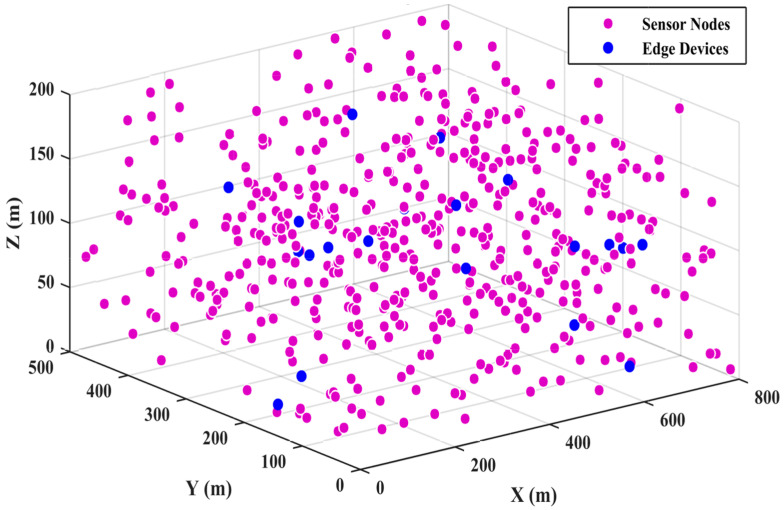
Sensor nodes and edge device deployment with 800 m × 500 m × 50 m for the proposed work.

**Figure 6 sensors-25-00286-f006:**
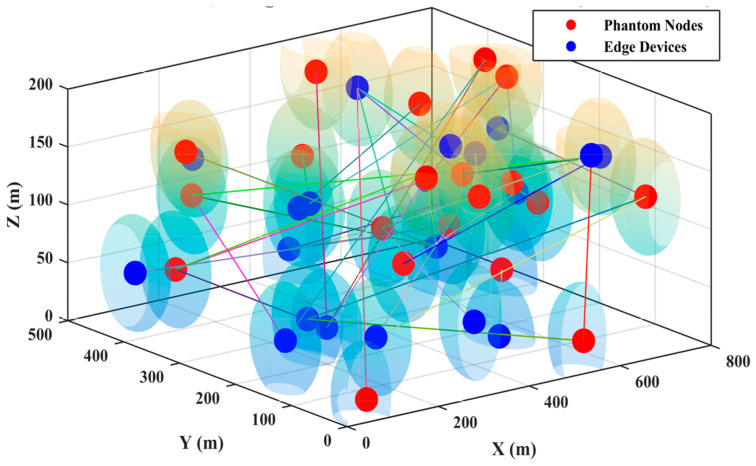
Deployment scenario of phantom node and edge devices with 800 m × 500 m × 200 m.

**Figure 7 sensors-25-00286-f007:**
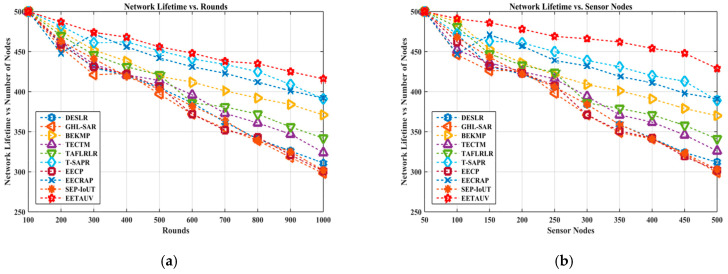
NetLT evaluation (**a**) against simulation rounds and (**b**) varying numbers of nodes.

**Figure 8 sensors-25-00286-f008:**
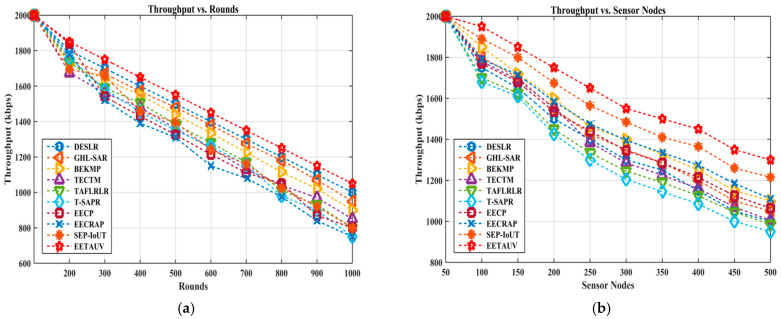
TPT evaluation (**a**) against simulation rounds and (**b**) varying numbers of nodes.

**Figure 9 sensors-25-00286-f009:**
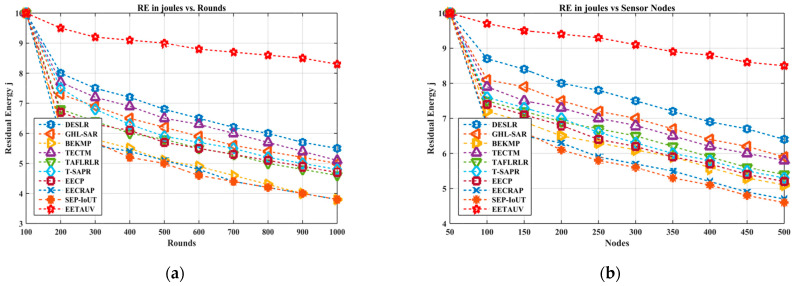
RE evaluation (**a**) against simulation rounds and (**b**) varying numbers of nodes.

**Figure 10 sensors-25-00286-f010:**
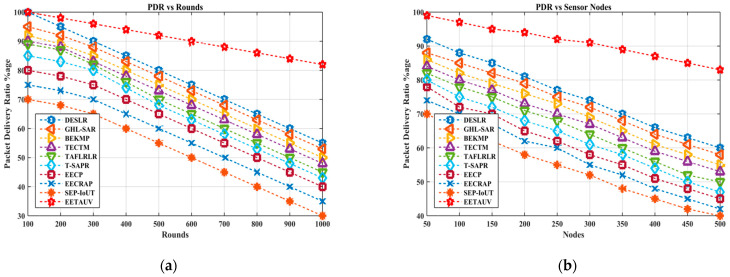
PDR evaluation (**a**) against simulation rounds and (**b**) varying numbers of nodes.

**Figure 11 sensors-25-00286-f011:**
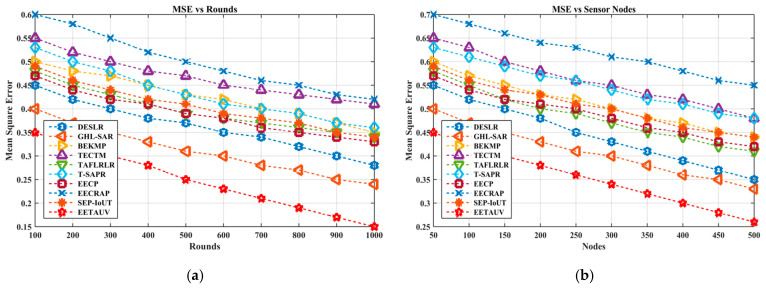
MSE evaluation (**a**) against simulation rounds and (**b**) varying numbers of nodes.

**Figure 12 sensors-25-00286-f012:**
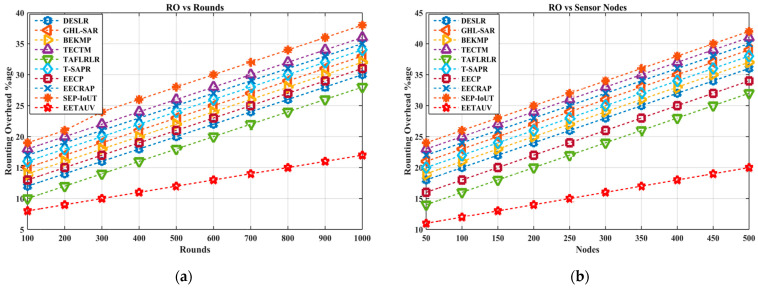
RO evaluation (**a**) against simulation rounds and (**b**) varying numbers of nodes.

**Figure 13 sensors-25-00286-f013:**
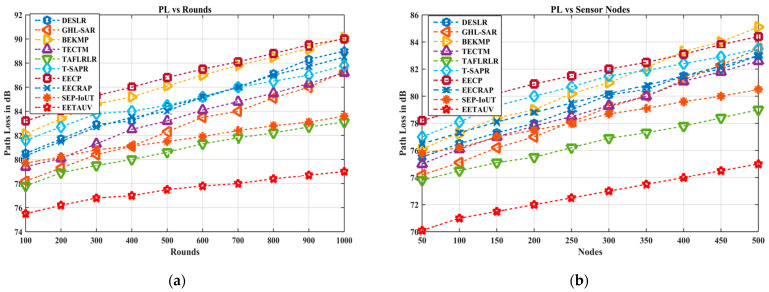
PL evaluation (**a**) against simulation rounds and (**b**) varying numbers of nodes.

**Figure 14 sensors-25-00286-f014:**
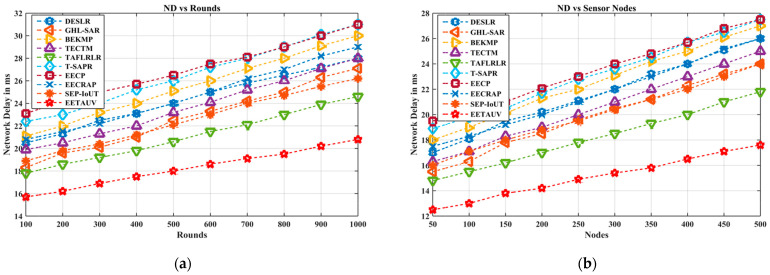
ND evaluation (**a**) against simulation rounds and (**b**) varying numbers of nodes.

**Figure 15 sensors-25-00286-f015:**
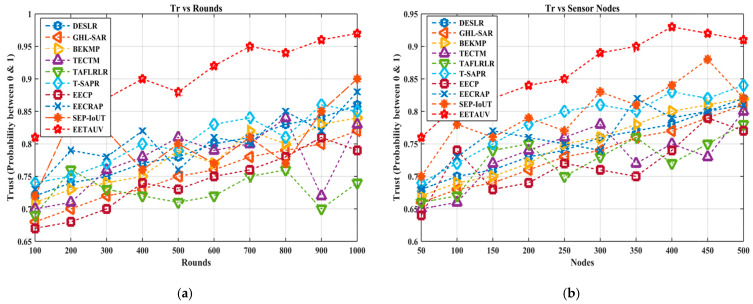
Trust evaluation (**a**) against simulation rounds and (**b**) varying numbers of nodes.

**Figure 16 sensors-25-00286-f016:**
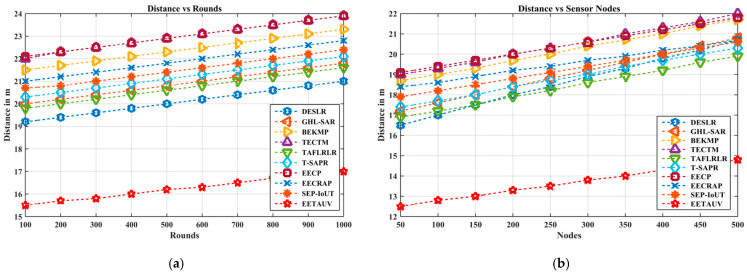
Distance evaluation (**a**) against simulation rounds and (**b**) varying numbers of nodes.

**Figure 17 sensors-25-00286-f017:**
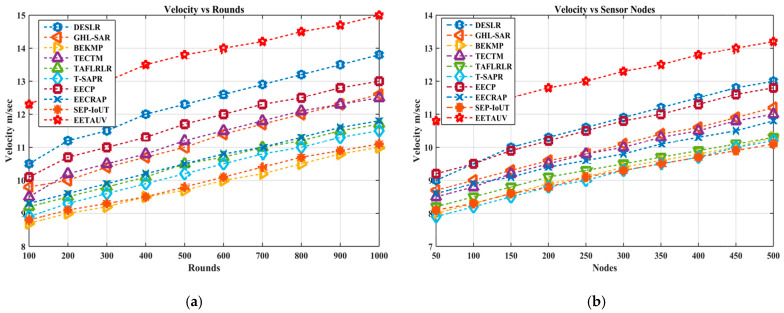
Velocity evaluation (**a**) against simulation rounds and (**b**) varying numbers of nodes.

**Figure 18 sensors-25-00286-f018:**
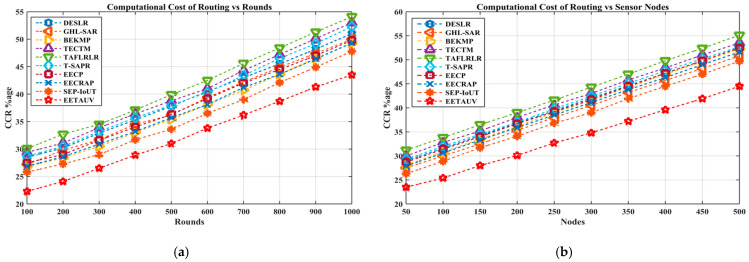
CCR evaluation (**a**) against simulation rounds and (**b**) varying numbers of nodes.

**Figure 19 sensors-25-00286-f019:**
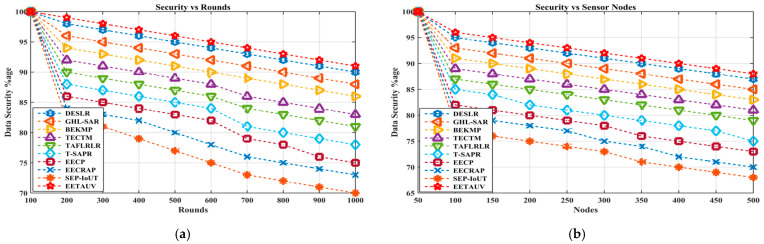
Data security evaluation (**a**) against simulation rounds and (**b**) varying numbers of nodes.

**Table 1 sensors-25-00286-t001:** Simulation parameters and their values.

Parameters	Values
Bandwidth	30 KHz
Data Bit Rate	1000–5000 bit-rate
Edge Devices	15
Maximum Energy	10 joules
Nodes Deployment	Randomly
Phantom Nodes	20 nodes
Sensor Nodes	100–500 SN
Simulation Area	800 m × 500 m × 50 m
Simulation Duration	8000 Rounds
Simulation Tool	MATLAB
Transmission Range of Single Node	50 m

**Table 2 sensors-25-00286-t002:** Cumulative performance of EETAUV vs. others.

Parameter	DESLR	GHL -SAR	BEKMP	TECTM	TAFLRLR	T-SAPR	EECP	EECRAP	SEP-IoUT	EETUAV (Proposed)
Network Lifetime	High	High	Medium	Medium	Medium	Low	High	High	Medium	Very High
Throughput	High	High	Medium	Medium	Low	Low	Low	Low	Medium	Very High
Residual Energy	Medium	Medium	Low	Medium	Low	Low	Low	Very Low	Very Low	Very High
PDR	Very High	High	High	Medium	Medium	Low	Low	Low	Low	Very High
Mean Square Error	Medium	Medium	High	High	Medium	Medium	Medium	High	High	Very Low
Routing Overhead	Medium	Medium	Medium	High	Medium	High	Medium	High	High	Very Low
Path Loss	Medium	Medium	Medium	Medium	High	Medium	Medium	Medium	High	Very Low
Network Delay	Medium	Medium	High	Medium	High	Medium	Medium	Medium	Medium	Very Low
Trust	Medium	Medium	Medium	Medium	Medium	Medium	Low	Low	Low	Very High
Distance	Low	Low	Medium	Medium	Medium	Medium	Medium	Medium	Medium	Very High
Velocity	Medium	Medium	Medium	Low	Low	Low	Low	Low	Low	Very High
Computational Cost of Routing	Medium	Medium	Medium	Medium	Medium	Medium	Medium	Medium	Medium	Very Low
Data Security	Low	Medium	Medium	Medium	Medium	Medium	Low	Low	Low	Very High

**Table 3 sensors-25-00286-t003:** Scale of values in terms of very low, low, medium, high, and very high.

Metric	Very Low	Low	Medium	High	Very High
NetLT Rounds	<400	400–410	410–420	420–430	>430
NetLT Nodes	<400	400–410	410–420	420–430	>430
TpT Rounds	<1300	1300–1350	1350–1400	1400–1450	>1450
TpT Nodes	<1300	1300–1350	1350–1400	1400–1450	>1450
RE Rounds	<5	5–6	6–7	7–8	>8
RE Nodes	<5	5–6	6–7	7–8	>8
PDR Rounds	<50	50–60	60–70	70–80	>80
PDR Nodes	<50	50–60	60–70	70–80	>80
MSE Rounds	>1	0.8–1	0.6–0.8	0.4–0.6	<0.4
MSE Nodes	>1	0.8–1	0.6–0.8	0.4–0.6	<0.4
RO Rounds	<10	10–20	20–25	25–30	>30
RO Nodes	<10	10–20	20–25	25–30	>30
PL Rounds	>90	85–90	80–85	75–80	<75
PL Nodes	>90	85–90	80–85	75–80	<75
ND Rounds	<20	20–25	25–30	30–35	>35
ND Nodes	<20	20–25	25–30	30–35	>35
Tr Rounds	<0.5	0.5–0.7	0.7–0.8	0.8–0.9	>0.9
Tr Nodes	<0.5	0.5–0.7	0.7–0.8	0.8–0.9	>0.9
D Rounds	<5	5–6	6–7	7–8	>8
D Nodes	<5	5–6	6–7	7–8	>8
V Rounds	<30	30–35	35–40	40–45	>45
V Nodes	<30	30–35	35–40	40–45	>45
CCR Rounds	<30	30–35	35–40	40–45	>45
CCR Nodes	<30	30–35	35–40	40–45	>45
Security Rounds	<70	70–75	75–80	80–85	>85
Security Nodes	<70	70–75	75–80	80–85	>85

**Table 4 sensors-25-00286-t004:** Summary of average values of each scheme in terms of simulation rounds.

Protocol	NetLT Rounds	TpT Rounds	RE Rounds	PDR Rounds	MSE Rounds	RO Rounds	PL Rounds	ND Rounds	Tr Rounds	D Rounds	V Rounds	CCR Rounds	Security Rounds
DESLR	394.1	1460.0	7.94	77.5	0.364	20	84.5	24.6	0.785	20.0	7.3	39.39	94.6
GHL-SAR	387.7	1432.5	6.8	75.1	0.302	25	81.4	23.2	0.758	21.0	7.5	37.66	92.8
BEKMP	424.4	1395.0	5.82	72.1	0.435	24	86.2	25.7	0.775	22.3	7.5	36.90	90.0
TECTM	403.2	1391.5	6.88	70.2	0.465	28	83.3	23.8	0.770	22.8	7.5	40.60	87.8
TAFLRLR	410.5	1340.5	6.52	67.9	0.396	18	80.6	20.7	0.731	20.5	7.2	40.32	85.0
T-SAPR	435.6	1319.0	6.77	65.5	0.440	24	84.7	26.7	0.798	21.0	7.4	40.02	82.8
EECP	390.7	1318.0	6.56	61.8	0.397	23	87.2	27.6	0.743	22.7	7.5	37.91	80.8
EECRAP	437.7	1289.5	5.62	56.8	0.504	25	84.3	24.5	0.80	21.6	7.6	37.79	78.0
SEP-IoUT	393.8	1335.0	5.72	53.8	0.397	31	81.7	22.1	0.804	23.3	7.1	34.73	76.6
EETAUV	454.7	1505.0	9.27	91	0.236	12.5	77.6	17.7	0.919	24.9	7.9	31.92	95.5

**Table 5 sensors-25-00286-t005:** Overall average values of each scheme in terms on varying number of nodes.

Protocol	NetLT Nodes	TpT Nodes	RE Nodes	PDR Nodes	MSE Node	RO Nodes	PL Nodes	ND Nodes	Tr Nodes	D Nodes	V Nodes	CCR Noes	Security Nodes
DESLR	395.2	1405.0	7.96	81.7	0.460	28	79.1	20.2	0.754	19.0	5.65	40.52	90.9
GHL-SAR	387.9	1456.0	7.49	75.9	0.404	32	78.2	19.5	0.724	19.9	5.55	39.40	89.1
BEKMP	424.7	1495.5	6.59	74.3	0.522	30	80.3	22.0	0.754	21.4	5.85	39.51	87.9
TECTM	402.2	1415.5	7.1	67.6	0.552	32.5	78.4	20.4	0.731	21.9	6.05	41.19	85.5
TAFLRLR	421.0	1369.5	6.99	66.8	0.487	22	76.6	17.4	0.73	19.7	6.05	42.40	80.7
T-SAPR	443.8	1414.0	6.94	61.0	0.546	30	81.0	23.0	0.759	20.0	6.15	40.51	78.9
EECP	391.0	1361.0	6.81	60.9	0.487	25	78.0	23.5	0.712	21.0	6.35	39.85	75.3
EECRAP	436.5	1378.5	6.54	59.7	0.623	30	80.2	21.2	0.773	19.8	5.65	39.30	75.6
SEP-IoUT	394.9	1411.5	6.42	57.2	0.520	34	79.6	19.0	0.787	21.3	6.05	37.80	72.4
EETAUV	468.3	1505.0	9.38	91.0	0.356	15.5	74.5	15.0	0.847	22.9	6.35	35.26	92.6

## Data Availability

Data are contained within the article.

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
