# Peer review of "Energy-Efficient and Trust-Based Autonomous Underwater Vehicle Scheme for 6G-Enabled Internet of Underwater Things"

_sensors, 2025, doi:10.3390/s25010286_

Round 1
Reviewer 1 Report
Comments and Suggestions for Authors
This paper proposes an energy-efficient and trust-based AUV scheme for 6G-enabled IoUT. The idea is novel and extensive simulations are conducted to verify the proposed method. There are some issues to be clarified and verified.
1. The contributions about the work summary have been list 8 tems, which are too redundant and should be summarized again concisely.
2. In the part of Literature Review, the relevant works are listed, but their comments are not analyzed. After reading this part, I have no ideas about the motivations about this work.
3. This paper should be reorganized. Is the EETUAV existing work? In part of Methodology, the authors list lots of formulas for energy consumption, distance and velocity, Network Delay and so on. The figure 3 and Algorithm 1 should be described in details instead of only physical quantities.
4. One idea of this paper is to use 6G nodes. Where the readers can see the advantages of 6G nodes instead of 5G or 4G or 3G?
5. The paper has many grammar issues such as
a) "The BS personnel's are connected..."
b) "Because of it large amount in advancement in computing and communication ideas..."
please check the whole paper to improve the readability of this paper.
6. The notations of equations are normative, such as:
a.In equation (10), 'n'?
b.For equation (16), where, d_max,v_max are bold.....Almost all equations have this kind of problem.
Please keep consistent between the equation and its denotation.
Comments on the Quality of English LanguageCan be further improved.
Author Response
Energy-Efficient and Trust-based Autonomous Underwater Vehicle Scheme for 6G-enabled Internet of Underwater Things
Dear Editor,
We appreciate the time and efforts of the editors and reviewers to provide their helpful feedback. We believe that all the insightful comments helped us to improve our manuscript. In the revised version, we have addressed all the reviewers’ concerns. For the editor's and reviewers’ convenience, newly added text and/or changes in the revised manuscript are highlighted in red color. The details of point-by-point responses to the reviewers' comments are presented in subsequent pages of this letter.
Sincerely,
The authors
Editor-in-Chief
Points to point comments are incorporated below.
Reviewer reports:
This paper proposes an energy-efficient and trust-based AUV scheme for 6G-enabled IoUT. The idea is novel and extensive simulations are conducted to verify the proposed method. There are some issues to be clarified and verified.
Comments 1: The contributions about the work summary have been list 8 tems, which are too redundant and should be summarized again concisely.
Response: Thank you, we deeply appreciate the valuable and insightful comments and suggestion of the reviewer to improve the quality of our article. We’ve updated and incorporated the suggested comments.
The primary contributions of the proposed EETAUV scheme are summarized as follows;
- Novel Energy-Efficient and Secure Communication Framework: We propose a 6G-enabled, Energy-Efficient and Trust-based Autonomous Underwater Vehicle (EETAUV) scheme for UASNs that integrates void node avoidance, localization techniques, and secure communication using normal and phantom nodes for node identification and verification.
- Enhanced Network Performance and Security: The scheme improves network stability, minimizes delay, increases packet delivery, and ensures secure data transmission with a lightweight, risk-aware strategy supported by AUVs for node discovery and verification.
- Comprehensive Evaluation: We extensively test the proposed EETAUV scheme against state-of-the-art methods using simulation metrics such as network lifetime, throughput, residual energy, packet delivery ratio, mean square error, routing overhead, path loss, network delay, trust, distance, velocity, computational cost, and data security, demonstrating superior cumulative performance.
Comments 2: In the part of Literature Review, the relevant works are listed, but their comments are not analyzed. After reading this part, I have no ideas about the motivations about this work.
Response: Thank You, We’ve provided issues with state of the art and also provided the major contributions of our proposed scheme.
The evolution of UASNs and IoUT has seen substantial progress, but several challenges persist in existing works. Energy consumption remains a critical bottleneck for prolonged network performance and operational stability, as highlighted by Mohamed et al. [3], who reviewed energy-efficient routing protocols in wireless sensor networks. While advancements in 5G-based technologies have introduced improvements, issues like high latency, limited bandwidth, and suboptimal routing strategies still hinder real-time underwater communication, as noted by Nkenyereye et al. [1]. Trust management models, extensively studied by Zhu et al. [11], Jiang et al. [12], and Du et al. [14], expose vulnerabilities in trust prediction accuracy and malicious attack identification, which compromise network security. Despite efforts to improve routing, protocols such as MO-CBACORP and SEECR introduced by Zhang et al. [19] and Saeed et al. [20] often fail to balance energy efficiency with reliable communication, particularly in highly dynamic underwater environments.
The integration of AUVs for data collection and mobility-enabled security has demonstrated potential; however, these systems struggle with void node issues, path reliability, and data transmission delays, as highlighted by Zhang et al. [7]. Emerging approaches leveraging machine learning and reinforcement learning, like those by Wang et al. [16] and Liu et al. [17], are computationally intensive, making them unsuitable for lightweight underwater networks. Additionally, advanced routing protocols such as DESLR and BEKMP, proposed by Zhu et al. [26] and Tomović et al. [28], incorporate layered routing and blockchain technologies but face challenges in scalability, adaptability, and sustained network longevity. Lastly, while approaches like the Depth-based Stable Election Routing Protocol by Ali et al. [34] focus on energy efficiency, they lack robust mechanisms for addressing heterogeneous IoUT systems and long-term stability in harsh underwater environments.
Research Gap and Major Contributions of the EETAUV Scheme
The proposed EETAUV scheme addresses the limitations of existing works by providing a comprehensive solution for UASNs. First, the EETAUV integrates a novel void node avoidance and localization mechanism enabled by 6G communication, ensuring real-time data transmission with minimal delays and improved energy efficiency. It effectively utilizes 6G's capabilities, such as ultra-low latency, high bandwidth, and enhanced connectivity, to optimize packet delivery while mitigating void node issues. Second, the scheme incorporates a trust-based communication model that secures data transmission through node identification and verification using phantom and normal nodes, thus enhancing network reliability and safeguarding against malicious attacks. This lightweight trust management strategy ensures secure communication while maintaining low computational overhead. Finally, the EETAUV scheme achieves superior network performance and stability by addressing critical metrics like energy consumption, network lifetime, packet delivery ratio, and routing overhead. Through extensive simulations, it demonstrates robust performance compared to state-of-the-art protocols, making it a scalable and adaptive solution for UASNs in dynamic underwater environments.
Comments 3: This paper should be reorganized. Is the EETUAV existing work? In part of Methodology, the authors list lots of formulas for energy consumption, distance and velocity, Network Delay and so on. The figure 3 and Algorithm 1 should be described in details instead of only physical quantities.
Response: We’ve updated the structure of the article accordingly as suggested. The EETUAV scheme is our proposed work and novel contribution; it’s not an existing work but in terms of its performance and evaluations, we’ve evaluated it with existing works.
Explanation of the Figure 3: The flowchart illustrates the step-by-step mechanism of data forwarding and routing in the EETAUV protocol to ensure energy-efficient and secure communication in 6G-enabled UASNs. The process begins with the initialization phase, where the route to the AUVR is established. This initialization leverages 6G communication to enable high-speed and low-latency data transfer, crucial for dynamic underwater environments. At this stage, the route cost (AUVR) is calculated to determine the most optimal communication path.
Next, the system integrates energy-efficient metrics to enhance the overall network lifetime by minimizing energy consumption during communication. This is critical for underwater networks, where energy resources are limited. The protocol then checks for void nodes—nodes that are either unreachable or have insufficient energy. If a void node is detected, the protocol switches to an alternate path and selects a neighboring route (AUVRj). The newly selected route undergoes a validity check to ensure it is suitable for forwarding data. If no valid node is found, the system updates the current route to maintain network stability.
When no void node is detected, the system transitions to the localization and 6G parameter phase, where the route cost is recalculated using advanced localization techniques and updated 6G communication parameters. These updates ensure that the routing path remains efficient and reliable. The protocol then evaluates the change in route cost (ΔAUVRC). If the change surpasses a predefined threshold, the system applies the Memphis criterion to verify the route based on 6G-specific thresholds. This step ensures that the routing path adheres to quality standards and performance expectations.
Finally, once all conditions are satisfied, the current route is updated to reflect the optimized routing path. This continuous process guarantees optimal network performance, energy efficiency, and reliable communication. The flowchart concludes when a stable route is achieved, or if further updates are unnecessary. This structured approach highlights the robustness and adaptability of the EETAUV protocol in addressing energy constraints and maintaining trust in underwater sensor networks.
Explanation of the Algorithm 1: The algorithm 1 details the inner workings of the EETAUV protocol. It starts by initializing system parameters such as the number of nodes, phantom nodes (for added security), and total energy available for each node. Each node is assigned initial attributes, including position and trust values, which facilitate secure and efficient routing. During the simulation, nodes dynamically update their distances and velocities to adapt to changing underwater environments. Void node detection plays a critical role in maintaining network robustness. If a node is identified as void, it switches to an alternate path to prevent communication breakdowns.
Energy consumption is carefully monitored throughout the process. The algorithm calculates the total energy consumed by all nodes, updating their trust values periodically to maintain secure communication. Localization energy is also calculated, accounting for the energy spent in determining node positions. Nodes' energy levels are subsequently updated to reflect their consumption. Network performance is evaluated using several key metrics, including network delay, PDR, path loss, routing overhead, MSE, and throughput. These metrics ensure the protocol meets performance standards while conserving energy.
The protocol stops if the total energy consumed exceeds the available energy, signaling the end of the simulation. Otherwise, it continues optimizing routing decisions and network operations. The final output includes performance metrics, providing a comprehensive evaluation of the protocol's effectiveness. Overall, the EETAUV protocol is robust and efficient, leveraging 6G-enabled features, trust-based mechanisms, and adaptive energy management to support secure and reliable underwater communication. It is particularly suited for dynamic underwater environments where energy efficiency, adaptability, and secure communication are paramount.
Comments 4: One idea of this paper is to use 6G nodes. Where the readers can see the advantages of 6G nodes instead of 5G or 4G or 3G?
Response: Thank you, we update the manuscript.
The transition from 3G, 4G, and 5G to 6G in implementing the EETAUV scheme for UASNs reflects a significant evolution in communication technology. Each previous generation faced specific limitations that constrained their effectiveness in underwater and high-demand network environments, particularly in terms of bandwidth, latency, energy efficiency, and adaptability. 3G networks, while revolutionary at their time, primarily focused on enabling basic mobile internet services and had limited bandwidth and high latency, making them unsuitable for handling the vast data exchange requirements and real-time communication demands of UASNs. 4G networks introduced higher speeds and better capacity; however, their coverage and energy efficiency still posed challenges for underwater networks, where signal attenuation and energy constraints are significant hurdles. Furthermore, 4G lacked the ultra-low latency and robust data-handling capabilities required for dynamic underwater scenarios. 5G, with its higher bandwidth, enhanced connectivity, and URLLC, made notable progress, particularly for terrestrial IoT and smart systems. However, even 5G struggled in underwater environments due to its limited ability to adapt to high levels of signal attenuation, dynamic mobility, and energy inefficiencies. Additionally, 5G networks still lacked the computational intelligence and optimization capabilities necessary for addressing the complex routing, localization, and trust management challenges posed by UASNs.
In contrast, 6G-enabled nodes address these issues and are uniquely suited for the EETAUV scheme due to their advanced technological features. 6G networks are designed to deliver terahertz (THz) frequencies, offering unprecedented bandwidth and ultra-low latency, which are crucial for supporting the high data rates and real-time communication required in underwater environments. This is particularly important for the EETAUV scheme, which relies on dynamic updates of node positions, trust values, and route costs. Moreover, 6G introduces energy-aware communication techniques, such as energy harvesting and adaptive power control, significantly extending the lifetime of underwater sensor nodes and addressing one of the critical limitations of earlier generations. Additionally, 6G incorporates artificial intelligence (AI) and machine learning (ML) frameworks, which enhance the routing and decision-making processes in EETAUV by providing predictive analytics, trust management, and adaptive routing strategies. This makes 6G more reliable in managing void node detection, route optimization, and energy consumption. Furthermore, 6G networks provide ubiquitous coverage, enabling seamless connectivity even in challenging underwater environments where traditional signals experience high path loss and interference. The integration of quantum communication and blockchain technologies in 6G enhances security and trust, addressing vulnerabilities in earlier networks, especially critical for ensuring secure data transmission in UASNs. Lastly, 6G's ability to handle extreme-scale IoT connectivity supports the high node density in UASNs, ensuring efficient communication and data forwarding even with hundreds of nodes. In short, while 3G, 4G, and 5G made incremental advancements, their limitations in bandwidth, latency, energy efficiency, and adaptability rendered them less suitable for the complex requirements of UASNs. 6G’s superior capabilities, such as ultra-wide bandwidth, AI-driven optimization, energy-efficient mechanisms, and enhanced security, make it the ideal choice for implementing the EETAUV scheme, ensuring optimal performance and reliability in underwater communication networks.
Comments 5: The paper has many grammar issues such as
- a)"The BS personnel's are connected..."
Response: Thank you, we update the manuscript and update the sentence. The offshore and onshore BSs are connected directly with the surface sink nodes to obtain meaningful information from the depth sensor nodes.
- b)"Because of it large amount in advancement in computing and communication ideas..."
Response: Thank you, we update the manuscript. Due to significant advancements in computing and communication technologies, there is now the potential to enable the deployment of the IoUV on ocean beds
Please check the whole paper to improve the readability of this paper.
Response: Thank you, We’ve properly checked the manuscript and all the math equations and corrected any typos or writing mistakes and grammatical mistakes.
Comments 6: The notations of equations are normative, such as:
a.In equation (10), 'n'?
Response: Thank you, we update the manuscript according. We’ve checked and corrected now Equation 10 accordingly as suggested.
b.For equation (16), where, d_max,v_max are bold.....Almost all equations have this kind of problem.
Response: Thank you, we update the formula.
Where, ​ = Maximum distance for localization, and ​ = Speed of sound in water.
Please keep consistent between the equation and its denotation.
Response: Thank you, We’ve properly checked all the math equations and corrected any typos or writing mistakes.

Reviewer 2 Report
Comments and Suggestions for Authors
-
Underwater Acoustic Sensor Networks (UASNs) are also known as Underwater Wireless Sensor Networks (UWSNs). Please note that they are not the same. UASN is a type of UWSN. UWSN can use acoustic, optical, or RF modes.
-
The authors have done a good job of conducting a literature survey of the work that has been done in terms of energy consumption to improve network performance and to enhance security and trust management to secure communication. However, it is important to identify the limitations of the studies presented and recognize the challenges that this paper aims to resolve. Please add a few lines on what the past studies are lacking.
-
The authors have presented great simulation results. However, the figures look stretched out which makes the fonts not on par with publication standards. Please update the Figures without stretching the fonts.
-
Please fix the following grammatical issues: We propose novel void node avoidance, localization 6g-enabled - Change to 6G.
-
To develop the autonomous system for IoUT is remaining a significant and challenging task - Please correct it to - Developing the autonomous system for IoUT remains a significant and challenging task.
-
Please use subscripts consistently. Eg: Ddata and Ccontrol(i) are not subscripted.
-
What are some of the drawbacks of your methods? What are some areas of improvement? Please add a short paragraph on that.
Please perform a thorough grammar and spell check.
Author Response
Energy-Efficient and Trust-based Autonomous Underwater Vehicle Scheme for 6G-enabled Internet of Underwater Things
Dear Editor,
We appreciate the time and efforts of the editors and reviewers to provide their helpful feedback. We believe that all the insightful comments helped us to improve our manuscript. In the revised version, we have addressed all the reviewers’ concerns. For the editor's and reviewers’ convenience, newly added text and/or changes in the revised manuscript are highlighted in red color. The details of point-by-point responses to the reviewers' comments are presented in subsequent pages of this letter.
Sincerely,
The authors
Editor-in-Chief
Points to Point comments are incorporated below.
Comments 1: Underwater Acoustic Sensor Networks (UASNs) are also known as Underwater Wireless Sensor Networks (UWSNs). Please note that they are not the same. UASN is a type of UWSN. UWSN can use acoustic, optical, or RF modes.
Response: Thank you, we update the manuscript.
Underwater Acoustic Sensor Networks (UASNs) are part of Underwater Wireless Sensor Networks (UWSNs) which are collection of multiple sensor nodes that operates in underwater region using the acoustic signals and gather data and other meaningful information.
Comments 2: The authors have done a good job of conducting a literature survey of the work that has been done in terms of energy consumption to improve network performance and to enhance security and trust management to secure communication. However, it is important to identify the limitations of the studies presented and recognize the challenges that this paper aims to resolve. Please add a few lines on what the past studies are lacking.
Response: Thank you,
The evolution of UASNs and IoUT has seen substantial progress, but several challenges persist in existing works. Energy consumption remains a critical bottleneck for prolonged network performance and operational stability, as highlighted by Mohamed et al. [3], who reviewed energy-efficient routing protocols in wireless sensor networks. While advancements in 5G-based technologies have introduced improvements, issues like high latency, limited bandwidth, and suboptimal routing strategies still hinder real-time underwater communication, as noted by Nkenyereye et al. [1]. Trust management models, extensively studied by Zhu et al. [11], Jiang et al. [12], and Du et al. [14], expose vulnerabilities in trust prediction accuracy and malicious attack identification, which compromise network security. Despite efforts to improve routing, protocols such as MO-CBACORP and SEECR introduced by Zhang et al. [19] and Saeed et al. [20] often fail to balance energy efficiency with reliable communication, particularly in highly dynamic underwater environments.
The integration of AUVs for data collection and mobility-enabled security has demonstrated potential; however, these systems struggle with void node issues, path reliability, and data transmission delays, as highlighted by Zhang et al. [7]. Emerging approaches leveraging machine learning and reinforcement learning, like those by Wang et al. [16] and Liu et al. [17], are computationally intensive, making them unsuitable for lightweight underwater networks. Additionally, advanced routing protocols such as DESLR and BEKMP, proposed by Zhu et al. [26] and Tomović et al. [28], incorporate layered routing and blockchain technologies but face challenges in scalability, adaptability, and sustained network longevity. Lastly, while approaches like the Depth-based Stable Election Routing Protocol by Ali et al. [34] focus on energy efficiency, they lack robust mechanisms for addressing heterogeneous IoUT systems and long-term stability in harsh underwater environments.
Research Gap and Major Contributions of the EETAUV Scheme
The proposed EETAUV scheme addresses the limitations of existing works by providing a comprehensive solution for UASNs. First, the EETAUV integrates a novel void node avoidance and localization mechanism enabled by 6G communication, ensuring real-time data transmission with minimal delays and improved energy efficiency. It effectively utilizes 6G's capabilities, such as ultra-low latency, high bandwidth, and enhanced connectivity, to optimize packet delivery while mitigating void node issues. Second, the scheme incorporates a trust-based communication model that secures data transmission through node identification and verification using phantom and normal nodes, thus enhancing network reliability and safeguarding against malicious attacks. This lightweight trust management strategy ensures secure communication while maintaining low computational overhead. Finally, the EETAUV scheme achieves superior network performance and stability by addressing critical metrics like energy consumption, network lifetime, packet delivery ratio, and routing overhead. Through extensive simulations, it demonstrates robust performance compared to state-of-the-art protocols, making it a scalable and adaptive solution for UASNs in dynamic underwater environments.
Comments 3: The authors have presented great simulation results. However, the figures look stretched out which makes the fonts not on par with publication standards. Please update the Figures without stretching the fonts.
Response: Thank you, we’ve updated all the figures accordingly and made sure that they’re not stretched out.
Comments 4: Please fix the following grammatical issues: We propose novel void node avoidance, localization 6g-enabled - Change to 6G.
Response: Thank you, we’ve changed the term 6g into 6G in the entire manuscript.
Comments 5: To develop the autonomous system for IoUT is remaining a significant and challenging task - Please correct it to - Developing the autonomous system for IoUT remains a significant and challenging task.
Response: Thank you, Developing the autonomous system for IoUT remains a significant and challenging task which includes computational intelligence decision and security.
Comments 6: Please use subscripts consistently. Eg: Ddata and Ccontrol(i) are not subscripted.
Response: Thank you, Where, Ccontrol​(i) represents the control messages generated by node i, and Ddata​(i) is the data transmitted by node i.
Comments 7: What are some of the drawbacks of your methods? What are some areas of improvement? Please add a short paragraph on that.
Response: Thank you for your insightful comment.
While the EETAUV scheme offers several advantages in terms of energy efficiency, trust management, and the integration of 6G communications for underwater acoustic sensor networks (UASNs), there are some potential drawbacks and areas for improvement. One limitation is the reliance on 6G communication, which, although promising, is still in the early stages of development, and its real-world deployment in underwater environments could face challenges related to infrastructure availability and high implementation costs. Additionally, while thes scheme effectively addresses energy consumption and trust management, the complexity of the algorithm may increase computational overhead, particularly in large-scale networks with numerous nodes, potentially leading to delays in real-time decision-making. Future work could focus on optimizing the routing algorithm for better scalability and reduced computational cost, as well as exploring hybrid communication models to balance energy efficiency and network reliability. Furthermore, incorporating more adaptive mechanisms for handling environmental changes, such as varying ocean currents and node mobility, could further enhance the robustness of the scheme in dynamic underwater environments.

Round 2
Reviewer 1 Report
Comments and Suggestions for Authors
The authors have solved my concerns.